# An Analysis of Concept Bottleneck Models: Measuring, Understanding, and Mitigating the Impact of Noisy Annotations

**Seonghwan Park, Jueun Mun, Donghyun Oh, Namhoon Lee**
POSTECH
{seonghwan.park,jueun1021,donghyunoh,namhoon.lee}@postech.ac.kr

## Abstract

Concept bottleneck models (CBMs) ensure interpretability by decomposing predictions into human interpretable concepts. Yet the annotations used for training CBMs that enable this transparency are often noisy, and the impact of such corruption is not well understood. In this study, we present the first systematic study of noise in CBMs and show that even moderate corruption simultaneously impairs prediction performance, interpretability, and the intervention effectiveness. Our analysis identifies a *susceptible* subset of concepts whose accuracy declines far more than the average gap between noisy and clean supervision and whose corruption accounts for most performance loss. To mitigate this vulnerability we propose a two-stage framework. During training, sharpness-aware minimization stabilizes the learning of noise-sensitive concepts. During inference, where clean labels are unavailable, we rank concepts by predictive entropy and correct only the most uncertain ones, using uncertainty as a proxy for susceptibility. Theoretical analysis and extensive ablations elucidate why sharpness-aware training confers robustness and why uncertainty reliably identifies susceptible concepts, providing a principled basis that preserves both interpretability and resilience in the presence of noise.

## 1 Introduction

Despite a decade of breakthroughs in deep learning, the inner-workings of end-to-end neural networks remain largely elusive [35, 24, 34]. This lack of transparency stems from their reliance on complex, high-dimensional internal representations, which offer limited insight into their decision-making process [37, 8]. Concept bottleneck models (CBMs) are designed to address this opacity by first predicting a set of human-interpretable *concepts*, which serve as an intermediate representation for deriving the final decision [19, 3, 38]. For instance, when it comes to the bird species classification problem [54], they first predict the concepts of the bird such as tail shape or body color, and then, infer the bird class based on these *attributes*, making the decision more understandable (*e.g.*, "this bird is HEERMANN GULL because it is gray and black in color with an orange beak!").

Formally, a CBM is trained on triplets of input $x \in \mathbb{R}^d$, binary concept vectors $c \in \{0,1\}^k$, and output $y$. The concept predictor $g : x \to c$ maps each input to its $k$-dimensional concept vector, and the output predictor $f : c \to y$ consumes those concepts to produce $y$ (see Figure 1a). This two-stage process offers a practical advantage, which is, domain experts can step in and edit the predicted concepts at inference, a process known as concept-level intervention [19]. For example, changing 'has stripe' from yes to no, while keeping other concepts such as 'number of legs', can shift the prediction from ZEBRA to HORSE without retraining and can recover much of the lost accuracy. Interventions further support counterfactual analysis [53], allowing users to ask "what if" scenarios

39th Conference on Neural Information Processing Systems (NeurIPS 2025).

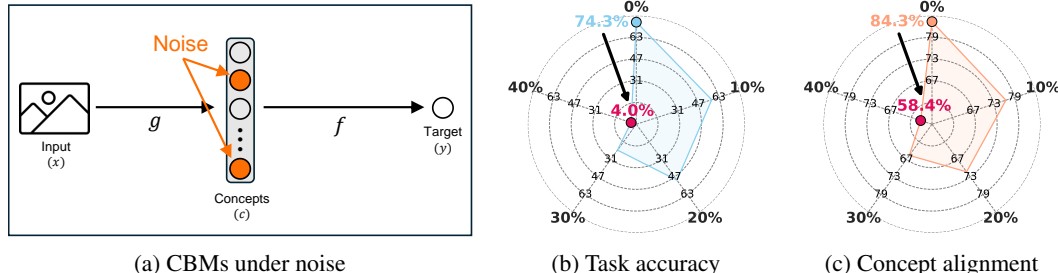

| (a) CBMs under noise | (b) Task accuracy | (c) Concept alignment |

Figure 1: (a) Noisy concept annotations arise easily in CBMs. (b, c) On the CUB [54] dataset, raising the noise level to 40% lowers task accuracy from 74.3% to 4.0% and reduces interpretability—assessed by the concept alignment score [61], which measures semantic alignment between learned representations and ground truth concepts—from 84.3% to 58.4%.

by manipulating concepts rather than raw pixels. Such human-in-the-loop interactivity is central to the appeal of CBMs and underscores the need to safeguard concept fidelity.

Despite their potential, we identify a critical issue inherent in CBMs: their reliance on concept annotations for training. This requirement entails extensive labeling efforts, and because of that, it can easily introduce annotation errors, *i.e.*, noisy concept labels. Unlike end-to-end models, such noise directly corrupts the intermediate concept bottleneck, the very foundation CBMs rely upon for interpretability and prediction, potentially rendering them more vulnerable to performance degradation under imperfect labels. Such noise can arise from human mistakes, subjective disagreements, and inconsistencies in annotator expertise, and yet, its impact on CBMs has been largely overlooked.

In this work, we provide the first systematic investigation into the impact of noisy annotations on CBMs, by comprehensively measuring its extent, understanding its underlying mechanisms, and mitigating its detrimental effects.

Specifically, our study reveals that noise significantly compromises key characteristics of CBMs, *i.e.*, interpretability, intervention effectiveness, as well as prediction performance (see Figure 1 for basic results). We discover that this degradation is primarily driven by a specific set of concepts that are highly *susceptible* to noise, whose accuracy drop largely accounts for overall performance declines. To mitigate these negative effects, we propose two strategies. First, we utilize sharpness-aware minimization (SAM) [10] during training to build robustness, particularly for noise-sensitive concepts. Second, observing that susceptible concepts often have high predictive uncertainty, quantified by entropy [41], we adopt an uncertainty-based intervention strategy at inference [21, 20, 43]. This strategy prioritizes correcting high-uncertainty concepts, enabling targeted interventions that significantly improve overall task performance. Extensive empirical results validate the effectiveness of this combined approach, consistently recovering the reliability of CBMs despite the presence of noise.

Our key contributions are summarized as follows:

- Section 2. We present the first comprehensive and systematic characterization of noise's impact on CBMs, demonstrating that even moderate level of noise significantly degrades predictive performance, interpretability, and intervention effectiveness.

- Section 3. To deepen our understanding of noise effects on CBMs, we identify a *susceptible set*, a small subset of concepts that suffer significant accuracy loss, and demonstrate that overall performance decline is largely driven by these localized failures.

- Section 4. To neutralize the risk of the susceptible set, we introduce two mitigation strategies: integrating sharpness minimization during training and implementing an uncertainty-based intervention at inference. Our suggestion is supported by theory and validated through extensive empirical evidence.

## 2 Measuring Impact

In this section, we systematically measure the impact of noise on CBMs across three key dimensions: prediction performance, interpretability, and intervention effectiveness. We begin by outlining our experimental setup (Section 2.1), followed by an evaluation of the noise effects on prediction performance (Section 2.2), interpretability (Section 2.3), and intervention effectiveness (Section 2.4).

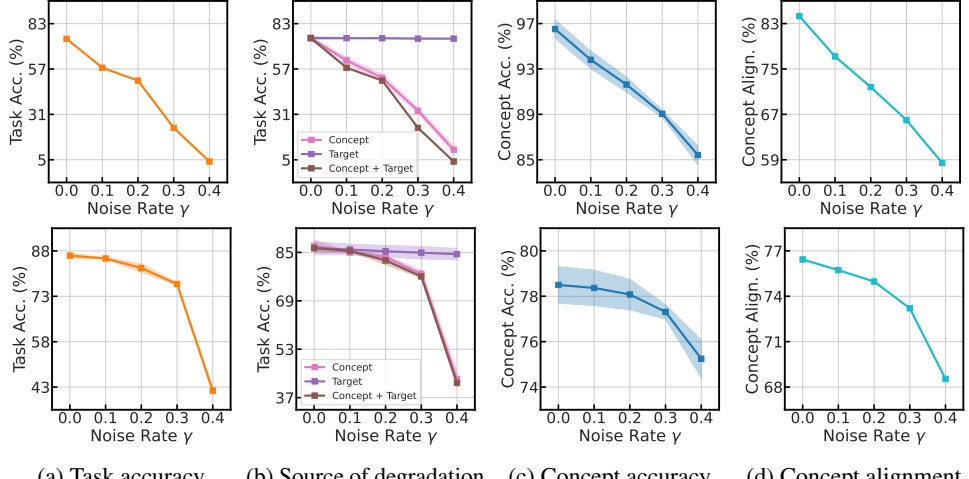

(a) Task accuracy    (b) Source of degradation    (c) Concept accuracy    (d) Concept alignment

Figure 2: Impact of noise on CBMs. (a) Task accuracy degradation; (b) Source of degradation; (c) Concept accuracy degradation; (d) Decline in interpretability measured by the concept alignment score. The top row represents the CUB dataset, and the bottom row shows results for AwA2 dataset.

## 2.1 Experimental Setup

Our experiments follow the settings proposed by Koh et al. [19], using InceptionV3 [47] as the concept predictor $g$ and a linear model as the target predictor $f$. We introduce noise into the CUB [54] and AwA2 [58] datasets to study noise impacts on CBMs. Specifically, each binary concept label $c_i \in \{0, 1\}$ is independently flipped with probability $\gamma$ to simulate concept noise $\widehat{c}$. For target noise $\widehat{y}$, each class label $y$ is randomly changed to a different label, uniformly sampled from the remaining classes, also with probability $\gamma$. This formulation of target noise is standard in the literature [31, 57, 26], and we adopt a parallel strategy for concept noise.

## 2.2 Prediction Performance

We begin by examining how noise impacts prediction performance. As shown in Figure 2a, increasing noise levels lead to a consistent decline in accuracy. For instance, introducing 30% noise into the AwA2 dataset results in a 9.4% drop in accuracy, while the same noise level in the CUB dataset leads to a dramatic 51% decline. Even modest noise has a surprisingly strong effect on CBMs, with just 10% noise in CUB causing a 16.6% drop in accuracy. To better un-

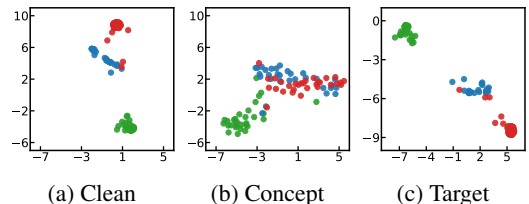

(a) Clean    (b) Concept    (c) Target

Figure 3: t-SNE [51] visualization of model representations under different noise settings.

derstand this vulnerability, we compare three types of noise: (i) concept-only noise, (ii) target-only noise, and (iii) combined noise (see Figure 2b). We find that concept noise alone leads to almost the same performance degradation as the combined case, suggesting that corrupted concept labels are the primary driver of performance loss. This conclusion is further supported by Figure 3, which visualizes internal representations using t-SNE [51] for three classes (*e.g.*, BLACK-FOOTED ALBATROSS (●), RED-WINGED BLACKBIRD (●), and YELLOW-HEADED BLACKBIRD (●)). Compared to clean supervision, concept noise severely distorts the class-wise clustering of embeddings, while target noise causes minimal disruption. These results demonstrate that concept-level corruption not only reduces prediction accuracy but also undermines the semantic structure of learned representations.

## 2.3 Interpretability

Next, we assess the effect of noise on interpretability. As shown in Figure 2c, concept prediction accuracy declines from 96.5% in clean conditions to 85.4% with 40% noise on the CUB dataset. Despite appearing modest, the 11.1% drop notably compresses the margin over random guessing in binary classification tasks, where minor concept prediction errors can compound and severely

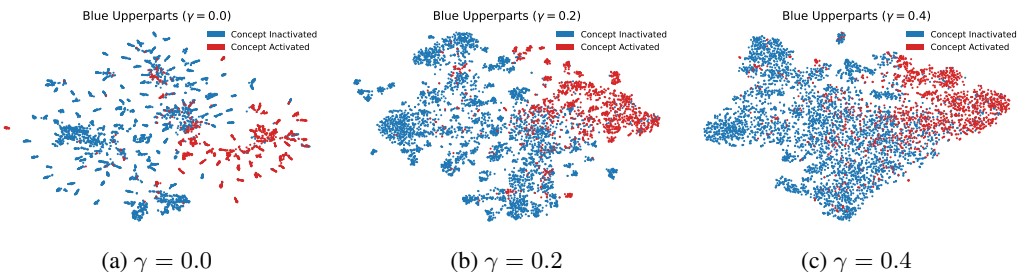

(a) $\gamma = 0.0$          (b) $\gamma = 0.2$          (c) $\gamma = 0.4$

Figure 4: t-SNE [51] visualizations of 'blue upperparts' concept embedding learnt in CUB with sample points colored red if the concept is active and blue if the concept is inactive in that sample. As noise ratio increases, the concepts are clearly entangled making concepts unreliable.

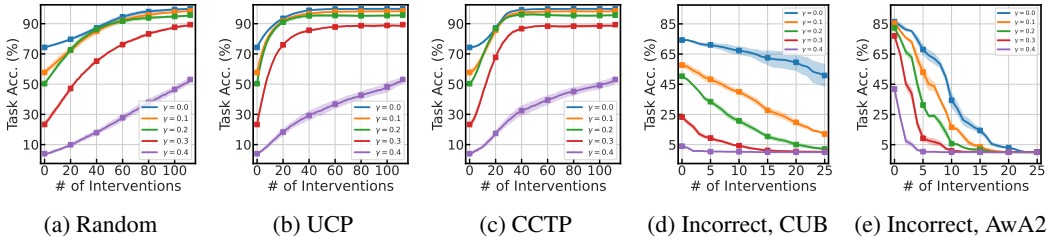

(a) Random     (b) UCP     (c) CCTP     (d) Incorrect, CUB     (e) Incorrect, AwA2

Figure 5: (a, b, c) Impact of noise on the effectiveness of CBM interventions using Random, UCP, and CCTP strategies under varying noise levels in the CUB dataset. (d, e) Effects of performing incorrect random concept interventions at different noise rates in CUB and AwA2 dataset.

impair target prediction in CBMs. Beyond accuracy, interpretability critically relies on semantic coherence of learned representations. Thus, we measure interpretability using the concept alignment score (CAS) [61], quantifying semantic alignment between learned representations and ground-truth concepts (see Figure 2d). We find CAS decreases substantially with increasing noise, *e.g.*, for each additional 10% increment of noise, CAS drops by approximately 7% on the CUB dataset, indicating severe degradation in interpretability.

Qualitative t-SNE [51] visualizations further illustrate this interpretability decline (see Figure 4). Clean embeddings for the specific concept 'blue upperparts' show distinct, semantically meaningful clusters corresponding to active (*i.e.*, red circles) and inactive (*i.e.*, blue circles) concept instances. However, increasing noise levels result in ambiguous, less separable clusters. Thus, noise compromises the explanatory power and reliability of concepts, diminishing interpretability. Additional examples are provided in Appendix D.12.

## 2.4 Intervention Effectiveness

Finally, we investigate how noise limits the utility of post-hoc interventions, utilizing three concept selection strategies proposed by Shin et al. [43]: (i) Random, which samples concepts uniformly [19]; (ii) UCP (uncertainty of concept prediction), which ranks concepts by predictive uncertainty; (iii) CCTP (contribution of concept on target prediction), which selects concepts exerting the greatest influence on the target prediction. Implementation specifics are detailed in Appendix C.5.

As shown in Figure 5a–5c, the benefits of intervention are increasingly constrained as noise levels rise (*i.e.*, the accuracy curves translate downward along the y-axis). Under the Random strategy, correcting every concept in a model trained with clean supervision yields near-optimal performance at 99.8% accuracy. However, this upper bound progressively declines to 98.4%, 95.7%, 89.4%, and 53.1% as the noise level increases from 10% to 40%. Notably, even exhaustive intervention at 40% noise fails to recover the performance of the clean model without any intervention. Similar trends are observed for the advanced intervention methods (*e.g.*, UCP and CCTP), indicating that the errors induced by noisy supervision in CBMs cannot be rectified through post-hoc correction alone.

To further evaluate robustness, we deliberately assign incorrect values to selected concepts, emulating the inevitable inaccuracies of expert intervention. As shown in Figure 5d–5e, CBMs trained on clean data tolerate a small number of such mistakes with only marginal performance decline. Under noise,

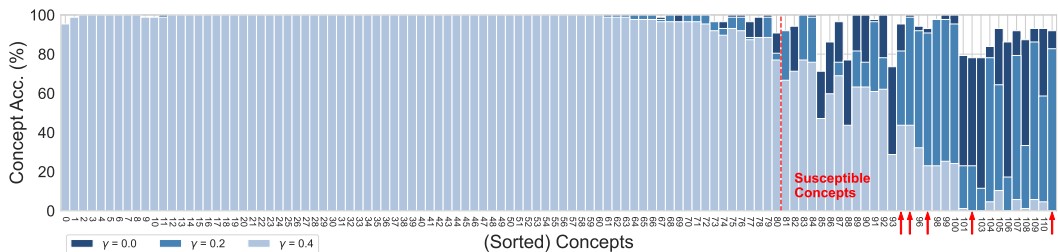

Figure 6: Impact of noise on individual concept accuracy. We evaluate the noise effects on concept predictions for LE CONTE SPARROW. As noise increases, accuracy declination becomes increasingly uneven across concepts. Concepts are sorted by their accuracy difference between 40% and 0% noise.

however, even slight intervention errors precipitate pronounced additional loss, revealing a fragility that limits the practical utility of concept-level correction in realistic human-expert workflows.

## 3 Understanding Mechanisms

In this section, we further analyze how noise affects the internal mechanisms of CBMs, separately examining the concept predictor $g$ and the target predictor $f$. We begin with the observation that noise induces non-uniform degradation of individual concept accuracies (Section 3.1). We then assess how noise disrupts the alignment between concepts representations and target predictions (Section 3.2).

### 3.1 Non-Uniform Degradation of Concept Accuracy

To understand the noise impact on CBMs, we analyze individual concept accuracy for a specific class, LE CONTE SPARROW, from the CUB dataset. We vary noise rates $\gamma \in \{0, 0.2, 0.4\}$ and measure accuracy across each concept.

As shown in Figure 6, although noise is introduced uniformly across all concepts, accuracy degradation is notably uneven. While the majority of concepts remain comparatively resilient, a distinct minority suffers a disproportionately large decline. We designate this minority as the *susceptible* concept set, defined as

$$S = \left\{ c_i \mid \Delta\mathrm{acc}_{c_i} > \overline{\Delta\mathrm{acc}} \right\}, \qquad (1)$$

where $\Delta\mathrm{acc}_{c_i} = \mathrm{acc}_{c_i}^{\mathrm{clean}} - \mathrm{acc}_{c_i}^{\mathrm{noisy}}$ represents the accuracy drop for concept $c_i$, and $\overline{\Delta\mathrm{acc}}$ denotes the mean drop across all 112 concepts. Under this criterion, approximately 23% of the concepts are identified as susceptible, suggesting heightened vulnerability to label corruption, likely rooted in greater semantic ambiguity.

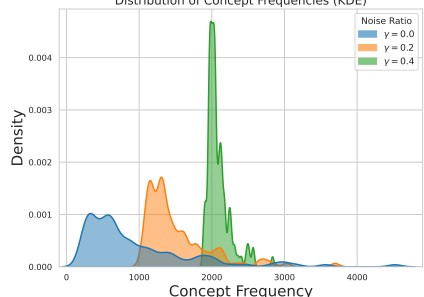

Figure 7: Concept frequency distribution (KDE). The distribution under noisy conditions (*e.g.*, 20%, 40%) differs substantially from that of the clean dataset.

To better understand this phenomenon, we plot the concept frequency distributions using kernel density estimation (KDE) [5] in Figure 7. In the clean dataset, concept occurrences are imbalanced, with many low-frequency and a few high-frequency concepts. As noise increases, however, this distribution becomes more uniform; at 40% noise, most concepts occur roughly 2,000 times. This shift may reduce the ability of model to learn rare but informative concepts, as the effective signal-to-noise ratio for such concepts deteriorates. Identifying and understanding these vulnerable concepts is essential, as their degradation can disproportionately affect downstream performance, which we explore in the next section.

### 3.2 Disruption in Representation Alignment

We next examine how noise during training reshapes the correspondence between learned representations and their associated target classes. In a noise-free setting we first identify the five most influential representation dimensions (*i.e.*, concepts) for the class LE CONTE SPARROW. These

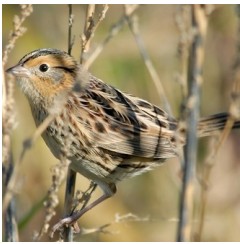

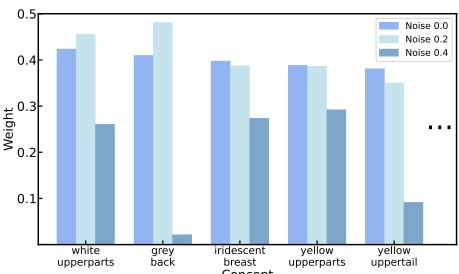

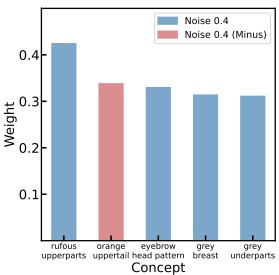

(a) Le Conte Sparrow          (b) Importance shift in key concepts          (c) Key concepts under 40% noise

Figure 8: Effect of noise on representation alignment. (b) Top 5 most influential (*i.e.*, key) concepts for LE CONTE SPARROW in a clean setting and tracks how their influence shifts as noise increases. (c) Top 5 influential concepts under 40% noise. In both figures, bars in blue denote positive weights, while red indicate negative weights, illustrating how noise reshapes concept importance.

dimensions align with meaningful visual attributes such as '`white upperparts`' and '`grey back`'. Their importance is quantified by the absolute magnitude of the weights in the linear target predictor $f$, a widely used proxy for feature saliency [25, 22, 6]. We then trace how these saliency values change as the proportion of corrupted labels increases.

As illustrated in Figure 8b, when the noise level is moderate (*i.e.*, 20%) the relative ordering of the predictive dimensions is largely preserved. At a higher noise level (*i.e.*, 40%), however, their saliency declines markedly. Informative dimensions lose influence, whereas irrelevant or spurious ones gain disproportionately large weights (see Figure 8c). Specifically, dimensions linked to '`grey back`' or '`yellow upper tail`' receive minimal weight, while a related attribute such as '`orange upper tail`' is assigned a negative weight. These shifts indicate a pronounced misalignment between the learned representations and their intended semantics.

We further relate this effect to concept-level susceptibility. In Figure 6, its five most influential dimensions, indicated with red arrows, coincide with the concepts most susceptible to noise. The same pattern emerges across the full dataset. Among 200 classes, 189 display an exact overlap between the five most influential dimensions and the noise-susceptible set. This widespread overlap exposes a critical weakness, *i.e.*, when concepts that are noise-sensitive coincide with distorted representations, performance degrades sharply. Thus, the intersection of susceptibility and disrupted alignment emerges as a principal source of instability, emphasizing the need for robust training methods to prevent representation collapse under noisy supervision.

## 4 Mitigating Effects

We have shown that noise severely degrades the key aspects of CBMs, primarily due to a subset of concepts (*i.e.*, *susceptible* concept set) that are especially vulnerable to noisy supervision and frequently coincide with representational misalignments. To address this, we introduce two mitigation strategies that operate at different stages of the model lifecycle: during training and inference. First, we incorporate sharpness-aware minimization (SAM) [10] at training time to improve the prediction accuracy of susceptible concepts (Section 4.1). Then, at inference time, we apply an uncertainty-based intervention strategy that prioritizes corrections on unreliable concept predictions (Section 4.2).

### 4.1 Training-Time Robustness via SAM

**Sharpness-aware minimization.** SAM induces parameters $\boldsymbol{w}$ that lie in flat regions of the loss landscape, yielding models that are less sensitive to perturbations and therefore generalize more reliably. Its objective is

$$\min_{\boldsymbol{w}} \max_{\|\boldsymbol{\varepsilon}\|_2 \leq \rho} \ell_{\text{train}}(\boldsymbol{w} + \boldsymbol{\varepsilon}), \tag{2}$$

where the inner maximization is approximated by $\widetilde{\varepsilon}(\boldsymbol{w}) = \rho \frac{\nabla_{\boldsymbol{w}} \ell_{\text{train}}(\boldsymbol{w})}{\|\nabla_{\boldsymbol{w}} \ell_{\text{train}}(\boldsymbol{w})\|_2}$. Gradient updates are then computed at $\boldsymbol{w} + \widetilde{\varepsilon}(\boldsymbol{w})$. By steering solutions toward flatter optima, SAM attenuates the performance degradation caused by noisy labels and other small perturbations [10, 2], rivaling the best dedicated noise-robust training methods [17, 62, 1].

Table 1: Comparison of concept accuracy, task accuracy, and CAS under different noise levels for BASE and SAM. SAM consistently enhances robustness across all evaluation metrics. Red values indicate the performance improvement achieved by SAM relative to BASE.

| METHOD | METRIC | CUB | | | AwA2 | | |
|---|---|---|---|---|---|---|---|
| | | $\gamma = 0.0$ | $\gamma = 0.2$ | $\gamma = 0.4$ | $\gamma = 0$ | $\gamma = 0.2$ | $\gamma = 0.4$ |
| BASE | CONCEPT ACC | $96.52\pm0.0$ | $91.63\pm0.0$ | $85.42\pm0.0$ | $78.50\pm0.8$ | $78.08\pm0.7$ | $75.25\pm0.8$ |
| | TASK ACC | $74.31\pm0.3$ | $50.35\pm0.7$ | $3.99\pm0.7$ | $86.45\pm0.9$ | $82.34\pm1.4$ | $41.85\pm1.0$ |
| | CAS | $84.31\pm0.0$ | $71.80\pm0.0$ | $58.44\pm0.0$ | $76.43\pm0.0$ | $74.97\pm0.0$ | $68.54\pm0.0$ |
| SAM | CONCEPT ACC | $97.19\pm0.1$ (+0.67) | $92.54\pm0.1$ (+0.91) | $86.31\pm0.1$ (+0.89) | $78.78\pm0.8$ (+0.28) | $78.47\pm0.5$ (+0.39) | $75.85\pm1.3$ (+0.60) |
| | TASK ACC | $78.96\pm0.8$ (+4.65) | $54.21\pm0.7$ (+3.86) | $4.95\pm1.4$ (+0.96) | $87.75\pm0.8$ (+1.30) | $85.73\pm0.4$ (+3.39) | $46.53\pm1.4$ (+4.68) |
| | CAS | $87.45\pm0.0$ (+3.14) | $74.18\pm0.0$ (+2.38) | $60.44\pm0.0$ (+2.00) | $77.74\pm0.0$ (+1.31) | $76.09\pm0.0$ (+1.12) | $70.10\pm0.0$ (+1.56) |

**SAM improves accuracy by stabilizing susceptible concepts.** We compare SAM-trained CBMs with SGD-trained ones (*i.e.*, BASE) across concept accuracy, task accuracy, and CAS on the CUB and AwA2 datasets under varying noise levels. As shown in Table 1, SAM consistently improves all metrics across noise conditions. Notably, even modest gains in concept accuracy lead to substantial improvements in downstream task performance. For in-

Table 2: Concept accuracy of *susceptible* and *non-susceptible* concepts. SAM improves accuracy in the noise-sensitive susceptible group. $\Delta$ indicates the gain of SAM over BASE for susceptible concepts.

| NOISE | OPTIMIZER | OVERALL | SUSCEPTIBLE | $\Delta$ | NON-SUSCEPTIBLE |
|---|---|---|---|---|---|
| $\gamma = 0.0$ | BASE | $96.52\pm0.0$ | - | | - |
| | SAM | $97.19\pm0.1$ | - | | - |
| $\gamma = 0.2$ | BASE | $91.63\pm0.0$ | $66.45\pm1.0$ | | $99.02\pm0.0$ |
| | SAM | $92.54\pm0.1$ | $70.30\pm0.7$ | **+3.85** | $98.93\pm0.1$ |
| $\gamma = 0.4$ | BASE | $85.42\pm0.0$ | $39.58\pm3.0$ | | $98.61\pm0.2$ |
| | SAM | $86.31\pm0.1$ | $43.65\pm1.2$ | **+4.07** | $98.64\pm0.3$ |

stance, under 40% noise in AwA2, a 0.60% increase in concept accuracy leads to a 4.68% gain in task accuracy. Previous work [2] attributed the robustness of SAM against noise to its $\ell_2$-regularization on final layer weights and intermediate activations. In Appendix A, we extend the theoretical analysis of Baek et al. [2] to CBMs, demonstrating that the same regularization effect holds.

We also emphasize that SAM is particularly effective at recovering concepts in the *susceptible* concept set. Table 2 shows that while concept accuracy in the non-susceptible set remains stable, SAM yields substantial gains in the susceptible set across all noise levels, *e.g.*, yielding gains of +3.85% and +4.07% under 20% and 40% noise, respectively. These findings underscore the effectiveness of SAM in selectively enhancing the learning of noise-sensitive concepts, thereby enhancing both predictive robustness and model interpretability under corrupted supervision.

## 4.2 Inference-Time Robustness via Uncertainty-Guided Intervention

**Intervention guided by susceptibility significantly enhances performance.** Although training with SAM effectively mitigates the adverse effects of noise, additional performance gains are achievable through targeted inference-time interventions. Motivated by our earlier findings that only a small subset of concepts significantly contributes to performance degradation, we prioritize corrections based on *susceptibility*, defined as the magnitude of accuracy decline under noise. Specifically, we select the five

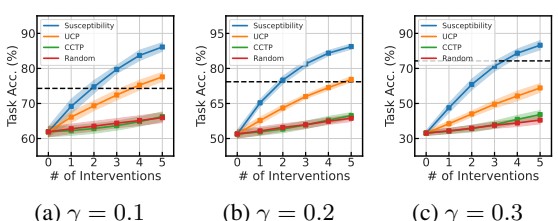

(a) $\gamma = 0.1$    (b) $\gamma = 0.2$    (c) $\gamma = 0.3$

Figure 9: Intervention results comparing baseline and susceptibility-based corrections. The dotted line indicates the clean-model accuracy without intervention.

most susceptible concepts per class for correction. As shown in Figure 9, susceptibility-guided corrections significantly outperform baseline interventions across varying noise levels. Remarkably, correcting even a single highly susceptible concept can yield accuracy improvements of around 10% at higher noise intensities (*e.g.*, 20% and 30%). Moreover, fewer than five targeted interventions frequently achieve or surpass the accuracy of models trained on clean data. These findings underscore that noise disproportionately affects a small subset of highly predictive and noise-sensitive concepts, and that targeted correction of these concepts substantially restores overall model performance.

**Susceptibility can be reliably approximated by uncertainty.** While susceptibility-based concept selection is a highly effective intervention strategy, it is not feasible to perform in realistic scenarios since it requires the access to a model trained with clean labels. To circumvent this challenge, we explore predictive uncertainty as a practical surrogate for susceptibility. Intuitively, concepts

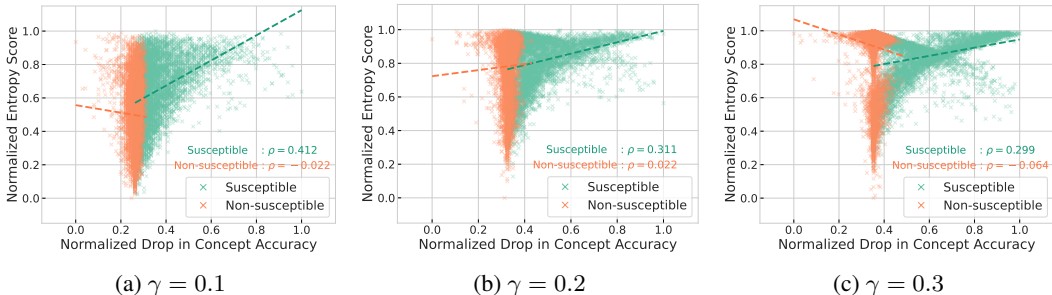

(a) $\gamma = 0.1$        (b) $\gamma = 0.2$        (c) $\gamma = 0.3$

Figure 10: Correlation between uncertainty and susceptibility. Scatter plots show predictive uncertainty (normalized entropy; y-axis) versus susceptibility (normalized accuracy drop; x-axis) across all concepts. Pearson correlation values ($\rho$) are reported separately for susceptible and non-susceptible sets, revealing a positive correlation in the former and no notable trend in the latter.

Table 3: Task accuracy on the CUB dataset under varying noise levels $\gamma$ and numbers of interventions $n$, comparing combinations of training strategies (BASE, SAM) and intervention methods (Random, Uncertainty-guided). The combination of SAM training and uncertainty-guided intervention consistently yields the highest accuracy, rapidly recovering performance with only a few interventions.

| | $\gamma = 0.0$ | | | $\gamma = 0.2$ | | | $\gamma = 0.4$ | | |
| METHOD | $n=0$ | $n=5$ | $n=10$ | $n=0$ | $n=5$ | $n=10$ | $n=0$ | $n=5$ | $n=10$ |
|---|---|---|---|---|---|---|---|---|---|
| BASE + RANDOM | $74.3\pm0.3$ | $75.6\pm0.4$ | $76.8\pm0.5$ | $50.3\pm0.6$ | $56.2\pm0.7$ | $62.1\pm1.0$ | $4.0\pm0.6$ | $4.8\pm0.7$ | $6.3\pm0.7$ |
| BASE + UNCERTAINTY | $74.3\pm0.3$ | $81.8\pm0.6$ | $87.0\pm0.7$ | $50.3\pm0.6$ | $71.2\pm1.7$ | $82.0\pm1.4$ | $4.0\pm0.6$ | $6.9\pm0.9$ | $10.7\pm1.4$ |
| SAM + RANDOM | $79.0\pm0.7$ | $80.1\pm0.6$ | $81.1\pm0.8$ | $54.2\pm0.6$ | $59.9\pm0.8$ | $65.6\pm0.9$ | $5.0\pm1.2$ | $5.6\pm0.8$ | $7.1\pm0.7$ |
| SAM + UNCERTAINTY | $79.0\pm0.7$ | $86.2\pm0.8$ | $90.9\pm0.7$ | $54.2\pm0.6$ | $75.2\pm0.7$ | $85.1\pm0.9$ | $5.0\pm1.2$ | $8.0\pm1.5$ | $12.4\pm1.7$ |

that contribute little to target prediction or are noise-sensitive should exhibit elevated uncertainty. To validate this hypothesis, we measure the Pearson correlation between concept-level uncertainty (quantified by the entropy of their predictive distributions [41]) and susceptibility. As depicted in Figure 10, we observe a positive correlation within the subset of susceptible concepts, whereas no consistent relationship emerges among non-susceptible concepts. These results suggest that predictive uncertainty effectively, though not definitively, reflects susceptibility induced by noise, supporting its use as a practical criterion for inference-time interventions. In Appendix B, we provide theoretical justification demonstrating that, under reasonable assumptions, uncertainty-based selection asymptotically approximates susceptibility-based concept selection.

**Uncertainty-based intervention markedly improves accuracy by targeting susceptible concepts.** Building on the observed correlation between predictive entropy and susceptibility, uncertainty provides a principled signal for inference-time interventions. Guided by the survey of Shin et al. [43], we employ uncertainty of concept prediction (UCP), which ranks concepts according to entropy $\mathcal{H}(\hat{c})$ [41] and prioritize corrections of the most uncertain ones. As illustrated in Figure 11, UCP consistently outperforms baseline methods such as Random selection and CCTP, with the margin of improvement increasing at higher noise intensities. These results indicate that uncertainty-based interventions effectively approximate susceptibility-based strategies, successfully identifying and rectifying noise-sensitive concepts.

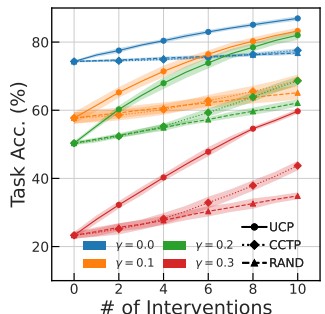

Figure 11: Task accuracy under different intervention strategies (*e.g.*, UCP, Random, CCTP).

Finally, we present our culminating experiment, examining the combined effects of training-time mitigation via SAM and inference-time corrections through UCP. Table 3 demonstrates that integrating SAM-based training with uncertainty-guided interventions nearly recovers the accuracy of cleanly trained models using only five targeted interventions at 20% noise. In stark contrast, models without SAM and using random corrections fall short even after ten interventions. Taken together, these findings underscore the complementary benefit of combining sharpness-aware training with uncertainty-driven interventions, substantially enhancing robustness and preserving interpretability under noisy supervision.

# 5 Discussion

**On the independence assumption**  Our primary objective is to provide a systematic analysis of how noise affects the performance and interpretability of CBMs. To isolate basic effects and to evaluate mitigation strategies in a controlled setting, we sought an experimental design in which concept noise and target noise can be manipulated independently. This design allows us to vary each factor without confounding and to attribute observed changes to a single source of corruption. For this reason, we considered the independence assumption a reasonable and transparent starting point for an initial analysis across a range of noise levels.

Nonetheless, we acknowledge that in real-world data, concept and target noise may be correlated. To account for this, we include additional experiments under more realistic conditions, simulating a correlated noise scenario where concept corruption occurs only when the target label is flipped, thereby inducing dependence between concept and target noise. The corresponding results are provided in Appendix D.4. In summary, even under dependent noise, the positive association between uncertainty and susceptibility persists for susceptible concepts. This pattern remains consistent across noise levels and indicates that our analysis and mitigation approach are effective in more realistic correlated noise settings. However, as the dependent noise considered here is still relatively simple, it is important to account for more complex correlation structures that may arise in real-world data.

**Benign appearance of target noise**  In our experiments, we observed that CBMs are highly sensitive to concept noise but relatively robust to target noise. This robustness can be attributed to the structure of the CBM, where the target predictor $f$, following prior work [19], is implemented as a single linear layer mapping concepts to the target. Its limited capacity makes it less able to overfit noisy target labels, reducing sensitivity to target corruption. Moreover, CBM training primarily burdens the concept predictor $g$. When $g$ produces stable, accurate concept representations, $f$ operates on relatively clean inputs, so even severe target noise has a muted effect on final accuracy.

**Generalization across CBM variants**  We primarily analyzed the impact of noise on CBMs using the most representative formulation, the original CBM [19]. To assess whether our methodology, particularly the application of SAM, generalizes to other CBM variants, we extend our analysis accordingly. Specifically, we evaluate several representative variants (*e.g.*, CEM [61], ECBM [59], AR-CBM [14], and SCBM [52]) to investigate how the presence of noise influences their performance. As summarized in Appendix D.2, all variants exhibit clear degradation in both concept and task accuracy as the noise level increases.

We further investigate whether the ability of SAM to mitigate the *susceptible sets* generalizes across different CBM variants. For SCBM, SAM showed clear improvements over SGD: at 20% noise, 75.8 vs. 74.9 (+0.9), and at 40% noise, 60.1 vs. 56.4 (+3.7). For AR-CBM and CEM, the gains were smaller: AR-CBM at 20% noise 71.9 vs. 71.3; at 40%, both 53.9; CEM at 20%, 73.3 vs. 73.1; at 40%, 54.6 vs. 54.5. These results indicate that the benefits of SAM vary across CBM variants and that resilience to susceptible concepts may depend on the underlying model design. While our focus remains on CBMs, we acknowledge that developing a broader and systematic framework encompassing multiple variants is important for understanding generalization under noisy conditions.

**Generalization across modalities**  CBMs are now being explored beyond vision, including settings that combine CBMs with large language models [49, 46]. The core CBM process first predicts concepts and then predicts the final target, which does not depend on a specific modality. Recent LLM based CBM studies employ a similar two stage design, so we expect our main findings to transfer to these settings. That said, concept annotated datasets for LLM are not yet systematically established. An important next step is to curate well-defined concept annotated datasets suitable for LLM applications and to analyze how label noise in such datasets affects CBMs.

**Rationale for employing SAM**  We adopted SAM based on prior findings by Baek et al. [2], which demonstrate that SAM is a representative and often effective approach for alleviating label noise, and it can outperform alternative methods [62, 1, 17]. In our experiments, SAM also mitigated the impact of noisy concept annotations in CBMs, particularly for *susceptible* concepts, confirming its suitability as a mitigation method in this setting. We do not claim that SAM is the only solution; rather, we present it as one effective approach to the observed problem and note that identifying other suitable mitigation methods for different CBM variants is also important.

# 6 Conclusion

**Further analyses** Beyond the primary experiments, we undertook a suite of supplementary studies to deepen our evaluation of noise robustness in CBMs. First, we performed ablation studies comparing alternative training strategies across varying noise levels. Second, we benchmarked our framework against representative noise-mitigation baselines. Third, we evaluated robustness on several advanced CBM variants under diverse noise configurations. Finally, we carried out auxiliary experiments probing orthogonal aspects of resilience. Comprehensive results are available in Appendix D.

**Comparisons** Prior work has pursued robustness in related settings, yet each tackles a distinct aspect. Sinha et al. [44] analyzed adversarial perturbations of continuous concepts to develop defenses against malicious attacks, Sheth and Ebrahimi Kahou [42] enforced disentangled representations via auxiliary losses to improve generalization under distribution shifts, and Penaloza et al. [30] proposed a preference-optimization objective to reduce noise sensitivity. In contrast, our study offers the first unified framework that measures, explains, and mitigates noise effects through the combined use of sharpness-aware training and uncertainty-guided correction. Theoretical analysis and extensive empirical evaluation substantiate the effectiveness of this unified approach.

**Limitations** Despite the substantial robustness gains provided by our framework, several limitations warrant further investigation. First, the method assumes well-calibrated uncertainty estimates, which is an assumption that may not hold in data-scarce or highly imbalanced regimes. Second, the reliance on a linear target predictor, while beneficial for interpretability, may hinder generalization on tasks with more complex decision boundaries. Third, our study is confined to binary concept labels; extending the framework to hierarchical, multi-class, or continuous concepts remains a promising avenue for future work. Addressing these limitations would further enhance the practical utility of CBMs in real-world environments characterized by noisy or incomplete supervision.

**Future work** Our research opens up new avenues for future research. First, extending our mitigation strategy to hierarchical, multi-class, or continuous concept spaces [32, 29] would enable applications such as fine-grained scene understanding and clinical severity grading. Second, the wider adoption of concept labels generated by large language models [60, 27, 45, 46] calls for a systematic study of their noise properties and their downstream impact on CBMs, which may offer important insights for both annotation protocols and model design. Third, incorporating semi-supervised objectives [16] and using active selection to identify the most informative concepts can decrease reliance on expensive or error-prone human labeling. Progress in these areas will further advance the deployment of robust and interpretable CBMs in real-world scenarios where supervision is noisy or incomplete.

**Closing remark** This analysis of CBMs establishes three core contributions as follows:

i. **Measuring impact** We demonstrate that even moderate noise markedly diminishes the central strengths of CBMs (*e.g.*, predictive performance, interpretability, and intervention efficacy) thereby revealing their vulnerability to imperfect supervision.

ii. **Understanding mechanisms** We disclose that a small subset of *susceptible* concepts can account for the majority of performance degradation. When these concepts coincide with misaligned representations in the target predictor, they trigger pronounced performance decline.

iii. **Mitigating effects** We propose a two-stage strategy: (i) sharpness-aware minimization during training, which stabilizes noise-sensitive concepts; and (ii) uncertainty-guided intervention at inference, which corrects only the most unreliable concept predictions. Both components are theoretically motivated and supported by extensive empirical evidence.

Collectively, these contributions establish a principled framework for building concept-based models that remain reliable and transparent in the presence of noisy supervision.

## Acknowledgement

This work was partly supported by the Institute of Information & communications Technology Planning & Evaluation (IITP) grants (RS-2019-II191906, Artificial Intelligence Graduate School Program (POSTECH), RS-2022-II220959, (part2) Few-Shot learning of Causal Inference in Vision and Language for Decision Making), and the National Research Foundation of Korea (NRF) grant funded by the Korea government (MSIT) (RS-2023-00210466).

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

# Contents

## A  Theoretical Analysis for Sharpness-Aware Minimization for CBMs

**Definition A.1** (2-Layer Concept Bottleneck Model (CBM)). *In a 2-layer* CBM, *the input $x \in \mathbb{R}^d$ is used to predict $k$ binary concepts $c = [c_1, c_2, \ldots, c_k]$, where $c_j \in \{0, 1\}$. The model consists of two layers. For each concept $c_j$, a shared first layer extracts intermediate activations, while a distinct second layer is specifically designed to predict that concept:*

- ***First layer:*** *$z = Wx$, where $W \in \mathbb{R}^{m \times d}$ is the weight matrix, and $z \in \mathbb{R}^m$ represents the intermediate activations.*
- ***Second layer:*** *$g(w_j, x) = \langle v_j, z \rangle = \langle v_j, Wx \rangle$, where $v_j \in \mathbb{R}^m$ is the weight vector associated with concept $c_j$.*

*The predicted probability for each concept $c_j$ is computed using the sigmoid activation:*

$$\sigma(g(w_j, x)) = \frac{1}{1 + e^{-g(w_j,x)}}. \tag{3}$$

**Definition A.2** (Binary Cross-Entropy (BCE) Loss). *The Binary Cross-Entropy (BCE) loss for a single concept $c_j$ is defined as:*

$$\ell(w_j, x, c_j) = -c_j \log(\sigma(g(w_j, x))) - (1 - c_j) \log(1 - \sigma(g(w_j, x))), \tag{4}$$

*where $g(w_j, x)$ is the model output for concept $c_j$. The total loss for all $k$ concepts is given by:*

$$\mathcal{L}(w, x, c) = \sum_{j=1}^{k} \ell(w_j, x, c_j). \tag{5}$$

**Definition A.3** (Sharpness-Aware Minimization (SAM)). SAM *optimizes the following objective to minimize the sharpness of the loss landscape:*

$$\min_{w} \max_{\|\varepsilon\|_2 \leq \rho} \mathcal{L}(w + \varepsilon, x, c), \tag{6}$$

*where $\varepsilon = [\varepsilon^{(1)}, \varepsilon^{(2)}]$ represents perturbations applied to $W$ and $v$ respectively, and $\rho$ controls the magnitude of the perturbation. For 2-layer CBM, $\varepsilon_j = [\varepsilon^{(1)}, \varepsilon_j^{(2)}]$ is applied to $W$ and $v_j$, respectively.*

Before starting the analysis, we define the following variants of SAM:

$$\text{1-SAM: } \nabla_{w+\varepsilon} \ell(w + \varepsilon, x, t) = t\sigma(-tf(w + \varepsilon, x))\nabla_{w+\varepsilon} f(w + \varepsilon_i, x)$$

$$\text{Jacobian SAM: } \Delta^{\text{J-SAM}} \ell(w + \varepsilon, x, t) = t\sigma(-tf(w, x))\nabla_{w+\varepsilon} f(w + \varepsilon, x).$$

Here, J-SAM applies the SAM perturbation only to the Jacobian term. Baek et al. [2] has demonstrated that the majority of the effectiveness of SAM is also reflected in J-SAM. Following the previous work, we analyze the efficacy of SAM in CBM using the J-SAM.

**Proposition A.4.** *In CBM with a 2-layer deep linear network $g(w_j, x) = \langle v_j, z \rangle = \langle v_j, Wx \rangle$, J-SAM introduces adaptive $\ell_2$-regularization on both the intermediate activations and the final-layer weights.*

*Proof.* The SAM update for the first-layer weight $W$ in CBM is given by:

$$-\nabla_{W+\varepsilon^{(1)}} \mathcal{L}(w + \varepsilon, x, c) = \sum_{j=1}^{k} -\nabla_{W+\varepsilon^{(1)}} \ell(w_j + \varepsilon, x, c_j) \tag{7}$$

$$= \sum_{j=1}^{k} -(\sigma(g(w_j, x)) - c_j)(v_j + \varepsilon_j^{(2)})x^\top \tag{8}$$

$$= \sum_{j=1}^{k} -(\sigma(g(w_j, x)) - c_j)(v_j + \rho\frac{(\sigma(g(w_j, x)) - c_j)z}{\|\nabla\mathcal{L}(w, x, c)\|_2})x^\top \tag{9}$$

$$= \sum_{j=1}^{k} -\nabla_W \ell(w_j, x, c_j) - \frac{\rho(\sigma(g(w_j, x)) - c_j)^2}{\|\nabla\mathcal{L}(w, x, c)\|_2}zx^\top \tag{10}$$

where $z = Wx$ is the intermediate activation, and the SAM update for the second-layer weight $v_j$ is given by:

$$-\nabla_{v_j + \varepsilon_j^{(2)}} \ell(w_j + \epsilon, x, c_j) = -(\sigma(g(w_j, x)) - c_j)(W + \varepsilon^{(1)})x \tag{11}$$

$$= -(\sigma(g(w_j, x)) - c_j)(W + \rho\frac{\sum_{i=1}^{k}(\sigma(g(w_i, x)) - c_i)v_i x^\top}{\|\nabla\mathcal{L}(w, x, c)\|_2})x \tag{12}$$

$$= -\nabla\ell(w_j, x, c_j) - \frac{\rho(\sigma(g(w_j, x)) - c_j)}{\|\nabla\mathcal{L}(w, x, c)\|_2} \sum_{i=1}^{k}(\sigma(g(w_i, x)) - c_i)\|x\|^2 v_i \tag{13}$$

$$\square$$

Proposition A.4 demonstrates that SAM introduces a penalty based on the intermediate activations $zx^\top = \nabla_W \frac{1}{w}\|z\|^2$, scaled by the squared prediction error, along with factors $\rho$ and a scalar factor. Additionally, SAM imposes a weight norm penalty on the final-layer weights $v_j = \nabla_{v_j} \frac{1}{2}\|v_j\|^2$, which is similarly scaled by the prediction error and a scalar factor. Following the original work, by regularizing the weight, SAM facilitates focused learning on clean data while mitigating the influence of noisy data by maintaining manageable loss levels for clean data and capping the loss growth for noisy data.

# B  Theoretical Justification for Uncertainty-Based Intervention

We present a theoretical justification for selecting intervention concepts based on predictive uncertainty in CBMs. Specifically, we show that uncertainty-based selection approximates the *susceptible* intervention set, *i.e.*, the set of concepts whose correction yields the greatest expected improvement in task performance.

Let $\widehat{c} = (\widehat{c}_1, \ldots, \widehat{c}_k)$ denote the vector of predicted concept values produced by a CBM trained on potentially noisy concept labels, and let $c^* = (c_1^*, \ldots, c_k^*)$ denote the corresponding ground-truth concept labels. Let $f$ be the target predictor and $\ell$ the task loss function.

**Definition B.1** (Susceptibility). *The susceptibility of concept $c_i$ is defined as the expected reduction in loss achieved by correcting the prediction $\widehat{c}_i$ to its ground-truth value $c_i^*$:*

$$\delta_i := \mathbb{E}_{(x,y)} \left[ \ell(f(\widehat{c}), y) - \ell(f(\widehat{c}_{-i}), y) \right], \tag{14}$$

*where $\widehat{c}_{-i}$ denotes the concept vector $\widehat{c}$ with the $i$-th entry replaced by $c_i^*$.*

Since $\widehat{c}$ encodes the consequences of training-time noise, $\delta_i$ quantifies how the model's noise-induced uncertainty at inference translates into potential performance gains through targeted correction.

**Definition B.2** (Predictive Uncertainty). *The predictive uncertainty of concept $c_i$ is quantified via Shannon entropy [41]:*

$$u_i := \mathcal{H}(\widehat{c}_i) = -\widehat{c}_i \log \widehat{c}_i - (1 - \widehat{c}_i) \log(1 - \widehat{c}_i). \tag{15}$$

**Definition B.3** (Concept Subsets). *Let $\delta = (\delta_1, \ldots, \delta_k)$ and $u = (u_1, \ldots, u_k)$ denote the vectors of susceptibility and uncertainty across all concepts. We define the top-$s$ subsets:*

$$\mathcal{S} := \mathrm{Top}_s(\delta) = \{i \in [k] \mid \delta_i \text{ ranks among the top-s in } \delta\}, \tag{16}$$

$$\mathcal{U} := \mathrm{Top}_s(u) = \{i \in [k] \mid u_i \text{ ranks among the top-s in } u\}. \tag{17}$$

Our objective is to characterize conditions under which the uncertainty-ranked set $\mathcal{U}$ approximates the true susceptibility-ranked set $\mathcal{S}$.

**Assumption B.4** (Linear Target Predictor and First-Order Approximation). *Assume the target predictor is linear: $f(\widehat{c}) = w^\top \widehat{c}$ for some $w \in \mathbb{R}^k$, and that $\ell$ is differentiable.*

*Applying a first-order Taylor expansion of $\ell(f(\widehat{c}_{-i}), y)$ around $\widehat{c}$ yields:*

$$\ell(f(\widehat{c}_{-i}), y) \approx \ell(f(\widehat{c}), y) + \nabla_{\widehat{c}} \ell(f(\widehat{c}), y)^\top (\widehat{c}_{-i} - \widehat{c}) \tag{18}$$

$$= \ell(f(\widehat{c}), y) + \frac{\partial \ell}{\partial \widehat{c}_i}(c_i^* - \widehat{c}_i), \tag{19}$$

*which implies:*

$$\delta_i \approx \mathbb{E}\left[\left| \frac{\partial \ell}{\partial \widehat{c}_i} \cdot (\widehat{c}_i - c_i^*) \right|\right]. \tag{20}$$

*By the chain rule:*

$$\frac{\partial \ell}{\partial \widehat{c}_i} = \frac{\partial \ell}{\partial f} \cdot \frac{\partial f}{\partial \widehat{c}_i} = \frac{\partial \ell}{\partial f} \cdot w_i, \tag{21}$$

*so:*

$$\delta_i \approx \mathbb{E}\left[\left| w_i \cdot \frac{\partial \ell}{\partial f} \cdot (\widehat{c}_i - c_i^*) \right|\right]. \tag{22}$$

**Assumption B.5** (Lipschitz Continuity and Boundedness of Loss Gradient). *Assume that the loss gradient $\frac{\partial \ell}{\partial f}$ is $L$-Lipschitz continuous:*

$$\left| \frac{\partial \ell}{\partial f}(f_1, y) - \frac{\partial \ell}{\partial f}(f_2, y) \right| \le L |f_1 - f_2| \quad \text{for all } f_1, f_2 \in \mathbb{R}. \tag{23}$$

*Moreover, suppose that the predicted value $f(\widehat{c})$ lies within a bounded interval $[a, b] \subset \mathbb{R}$. Then, for any reference point $f_0 \in [a, b]$, it follows from Lipschitz continuity that:*

$$\left| \frac{\partial \ell}{\partial f}(f(\widehat{c}), y) \right| \le \left| \frac{\partial \ell}{\partial f}(f_0, y) \right| + L \cdot |f(\widehat{c}) - f_0| \le M, \tag{24}$$

*for some constant $M$. This ensures that the gradient magnitude is uniformly bounded over the domain of interest.*

**Simplification.** To isolate the effect of uncertainty, we assume $|w_i|$ is approximately constant across all concepts. Under this simplification and Assumption B.5:

$$\delta_i \propto \mathbb{E}[|\widehat{c}_i - c_i^*|]. \tag{25}$$

**Assumption B.6** (Linear Relationship Between Uncertainty and Prediction Error). *In binary classification, assume $\widehat{c}_i \in (0,1)$ and $c_i^* \in \{0,1\}$. The expected absolute error is given by:*

$$\mathbb{E}[|\widehat{c}_i - c_i^*|] = 2\widehat{c}_i(1 - \widehat{c}_i). \tag{26}$$

*Observe that both $2\widehat{c}_i(1 - \widehat{c}_i)$ and the entropy $\mathcal{H}(\widehat{c}_i)$ are symmetric about $\widehat{c}_i = 0.5$ and attain their maximum at this point. Assume that the expected absolute error is approximately proportional to the predictive uncertainty:*

$$\mathbb{E}[|\widehat{c}_i - c_i^*|] \approx \gamma \cdot u_i \quad \text{for some } \gamma > 0, \tag{27}$$

*where $u_i := \mathcal{H}(\widehat{c}_i)$ denotes the entropy-based uncertainty of concept $c_i$.*

*Under this approximation, the susceptibility admits the following linear-noise model:*

$$\delta_i = \alpha u_i + \varepsilon_i, \tag{28}$$

*where $\alpha > 0$ and $\varepsilon_i$ is a zero-mean noise term with finite variance.*

**Definition B.7** (Kendall's Tau Distance). *Given two rankings $\pi_\delta$ and $\pi_u$ of $k$ items, the Kendall's Tau distance counts the number of discordant pairs:*

$$\tau(\pi_\delta, \pi_u) := |\{(i,j) \mid i < j, \ (\pi_\delta(i) - \pi_\delta(j)) \cdot (\pi_u(i) - \pi_u(j)) < 0\}|. \tag{29}$$

**Lemma B.8** (Ranking Consistency via Kendall's Tau). *Suppose $\delta_i = \alpha u_i + \varepsilon_i$, where the $\varepsilon_i$ are independent, zero-mean, and have bounded variance. Let $\pi_\delta$ and $\pi_u$ denote the rankings induced by $\delta$ and $u$, respectively. Then as $\sigma^2 \to 0$:*

$$\mathbb{E}[\tau(\pi_\delta, \pi_u)] \to 0. \tag{30}$$

*Proof.* For any pair $(i,j)$, we can write:

$$\delta_i - \delta_j = \alpha(u_i - u_j) + (\varepsilon_i - \varepsilon_j). \tag{31}$$

As $\sigma^2 \to 0$, the perturbation term $\varepsilon_i - \varepsilon_j$ vanishes in probability. Therefore,

$$\mathbb{P}\left[\text{sign}(\delta_i - \delta_j) \neq \text{sign}(u_i - u_j)\right] \to 0. \tag{32}$$

Summing over all $\binom{k}{2}$ concept pairs establishes the result. $\square$

**Proposition B.9** (Approximate Recovery of the Susceptible Set). *Under Assumption B.4 and B.5, and assuming the linear-noise model (28), the uncertainty-based top-$s$ set $\mathcal{U}$ recovers the susceptibility-based set $\mathcal{S}$ in the low-noise regime:*

$$\lim_{\sigma^2 \to 0} \mathbb{P}[\mathcal{U} = \mathcal{S}] = 1. \tag{33}$$

*Proof.* By Lemma B.8, the rankings $\pi_u$ and $\pi_\delta$ converge in probability as the noise variance $\sigma^2$ tends to zero. Since both $\mathcal{U}$ and $\mathcal{S}$ are determined by the top-$s$ indices in their respective rankings, their equality follows in the limit. $\square$

This analysis provides both intuitive and formal support for the use of uncertainty-guided interventions in CBMs. While the assumptions (*e.g.*, constant weight or the approximate linear relationship between entropy and prediction error) may not hold strictly in practice, empirical evidence (see Figure 10) demonstrates a consistent and positive correlation between predictive uncertainty and susceptibility, validating the practical utility of this approach.

## C   Experimental Details

This section presents detailed information on datasets, CBM training strategies, implementation settings, label noise injection, and concept selection criteria. Some content may overlap with the main paper.

## C.1 Datasets

This study employs two real-world datasets that are widely used in prior work for evaluating the performance of CBMs under diverse conditions. CUB [54] is a fine-grained bird classification dataset consisting of 5,994 training and 5,794 test images. Following Koh et al. [19], we utilize 112 of the 312 most frequently occurring binary attributes for concept supervision [19, 61, 59]. AwA2 [58] is a benchmark for zero-shot learning comprising 37,322 images spanning 50 animal categories, each annotated with 85 binary attributes used as concept labels [59]. For both datasets, we adopt the preprocessing strategy described in Koh et al. [19], applying random color jitter, horizontal flipping, and cropping to a resolution of $224 \times 224$ during training. At inference time, images are center-cropped and resized to the same resolution.

## C.2 CBM Training Strategies

Following Koh et al. [19], we consider three training strategies for the concept predictor $g$ and the target predictor $f$: (i) *Independent*: The two components are trained separately. The target predictor $\widehat{f}$ is trained using the ground-truth concepts $c$ via

$$\widehat{f} = \operatorname*{argmin}_{f} \sum_i \ell_Y(f(c^{(i)}), y^{(i)}), \tag{34}$$

while the concept predictor $\widehat{g}$ is learned independently as

$$\widehat{g} = \operatorname*{argmin}_{g} \sum_{i,j} \ell_{C_j}(g_j(x^{(i)}), c_j^{(i)}). \tag{35}$$

At inference time, the final prediction is obtained by composing $\widehat{f}(\widehat{g}(x))$; (ii) *Sequential*: This strategy first trains the concept predictor $\widehat{g}$ as above, and subsequently optimizes the target predictor $f$ using the predicted concepts $\widehat{g}(x)$; (iii) *Joint*: The concept and target predictors are trained simultaneously by minimizing a combined loss:

$$\widehat{f}, \widehat{g} = \operatorname*{argmin}_{f,g} \sum_i \left[ \ell_Y(f(g(x^{(i)})), y^{(i)}) + \sum_j \lambda \ell_{C_j}(g(x^{(i)}), c^{(i)}) \right]. \tag{36}$$

Here, $\ell_{C_j} : \mathbb{R} \times \mathbb{R} \to \mathbb{R}_+$ denotes the loss function for the $j$-th concept, and $\ell_Y : \mathbb{R} \times \mathbb{R} \to \mathbb{R}_+$ denotes the loss for target prediction. Our main experiments are conducted using the *Independent* strategy, and comparative results for the other strategies are provided in Appendix D.1.

## C.3 Implementation Details

For concept prediction, we fine-tune InceptionV3 [47], pre-trained on ImageNet [7], on the CUB [54] and AwA2 [58] datasets. The target predictor $f$ is implemented as a single-layer linear classifier, following the standard setup in Koh et al. [19]. All experiments are conducted with three random seeds, and we report the average performance. Training and evaluation are conducted on NVIDIA GeForce RTX 3090 (24GB), RTX A6000 (48GB), and A100 (80GB) GPUs. The code is available at https://github.com/LOG-postech/CBM-Noise.

For the *independent* and *sequential* strategies, we use a learning rate of $10^{-2}$; the *joint* model is trained with a learning rate of $10^{-3}$. All models are trained for 500 epochs using a batch size of 64 and momentum of 0.9. We apply a weight decay of $4 \times 10^{-5}$ and reduce the learning rate by a factor of 0.1 when the validation loss does not improve for 10 consecutive epochs. Early stopping is triggered after 15 epochs. To address class imbalance in concept labels, we adopt a weighted cross-entropy loss as in Koh et al. [19]. For models trained with the SAM optimizer, the sharpness parameter $\rho$ is selected via grid search over $\{0.01, 0.05, 0.1\}$, while all other training hyper-parameters are kept consistent with the baseline. The optimal $\rho$ is selected based on task accuracy.

All additional CBM variants shown in Appendix D.2 are trained on the CUB dataset for 300 epochs using the Adam optimizer [18] with a fixed learning rate of $10^{-4}$, following Vandenhirtz et al. [52]. For the independently trained autoregressive CBM [14], training is scheduled such that two-thirds of the total epochs are allocated to learning the concept predictor $g$, and the remaining one-third to the target predictor $f$, consistent with Vandenhirtz et al. [52]. We employ ResNet-18 [15] as the

backbone for CEM, SCBM, and autoregressive CBM, and ResNet-101 for ECBM. The batch size is set to 256 for CEM, SCBM, and autoregressive CBM models, and 64 for ECBM. When applying SAM to these models, we perform a grid search over $\rho \in \{0.01, 0.05, 0.1\}$, while keeping all other training hyper-parameters aligned with those used for the Adam baseline.

We conduct a targeted hyperparameter search for each label noise mitigation technique evaluated in Appendix D.5, including label smoothing regularization (LSR) [47], symmetric cross-entropy learning (SL) [56], and SAM. For each method, we tune the corresponding parameters (see Table 4), such as the smoothing factor $\epsilon$ for LSR, regularization coefficients $\alpha$ and $\beta$ for SL, and the sharpness radius $\rho$ for SAM, under varying levels of noise ($\gamma \in \{0.0, 0.1, 0.2, 0.3, 0.4\}$). The best hyperparameters are selected based on task accuracy.

Table 4: Hyperparameter search ranges for label noise mitigation methods. Here, $g$ and $f$ refer to the concept predictor and target predictor in the CBM framework, respectively.

| Method | Hyper-parameter | Search Range | Model | Assignment | | | |
|---|---|---|---|---|---|---|---|
| | | | | $\gamma = 0.0$ | $\gamma = 0.1$ | $\gamma = 0.2$ | $\gamma = 0.3$ |
| LSR | initial LR | {0.01, 0.001, 0.0001} | g | 0.0001 | 0.0001 | 0.001 | 0.0001 |
| | | | f | 0.01 | 0.01 | 0.01 | 0.01 |
| | epsilon | {0.1, 0.2} | g | 0.1 | 0.1 | 0.2 | 0.2 |
| | | | f | 0.1 | 0.2 | 0.2 | 0.2 |
| SL | initial LR | {0.01, 0.001, 0.0001} | g | 0.001 | 0.001 | 0.0001 | 0.01 |
| | | | f | 0.01 | 0.01 | 0.01 | 0.001 |
| | alpha | {0.1, 1.0, 5.0} | g | 1.0 | 0.1 | 5.0 | 0.1 |
| | | | f | 5.0 | 5.0 | 0.1 | 1.0 |
| | beta | {0.1, 1.0} | g | 0.1 | 0.1 | 0.1 | 1.0 |
| | | | f | 0.1 | 0.1 | 1.0 | 1.0 |
| SAM | initial LR | {0.01, 0.001, 0.0001} | g | 0.0001 | 0.0001 | 0.0001 | 0.0001 |
| | | | f | 0.01 | 0.01 | 0.01 | 0.001 |
| | rho | {0.1, 0.2} | g | 0.2 | 0.2 | 0.1 | 0.2 |
| | | | f | 0.2 | 0.2 | 0.1 | 0.1 |

## C.4 Label Noise Injection

To simulate real-world annotation errors, we introduce synthetic label noise following two widely used paradigms: symmetric noise and asymmetric noise [23, 50, 33, 13, 55, 11]. Additionally, we consider grouped noise for concept corruption. These settings enable a systematic evaluation of CBM robustness under varying noise conditions.

   i. **Symmetric noise.** Each class label is randomly flipped to one of the remaining $N-1$ classes with uniform probability. Specifically, for a dataset with $N$ classes, the probability of mislabeling a sample as any incorrect class is $\frac{1}{N-1}$. This simulates class-independent, uniformly distributed noise. An analogous strategy is applied to concepts, where each concept label is flipped uniformly to a different one.

   ii. **Asymmetric noise.** Labels are deterministically flipped to adjacent classes in a cyclic manner. Specifically, each class label $i$ is reassigned to $(i+1) \bmod N$, simulating structured noise patterns often observed in ordinal or semantically neighboring categories. For concept labels, a similar procedure is applied: the label of the $i$-th concept is flipped, and the label of the $\big((i+1) \bmod K\big)$-th concept is also perturbed, thereby modeling concept-level misannotations between adjacent semantic dimensions.

   iii. **Grouped noise.** Concept labels are randomly corrupted within predefined semantically coherent groups (e.g., "wing color"), representing annotation inconsistencies typically arising in domain-specific settings.

Symmetric noise captures general annotation uncertainty, asymmetric noise models systematic labeling biases, and grouped noise reflects domain-driven confusions within related concept groups. Together, these diverse noise settings constitute a comprehensive testbed for assessing the robustness of concept-based models. Experimental results under these conditions are presented in Appendix D.4.

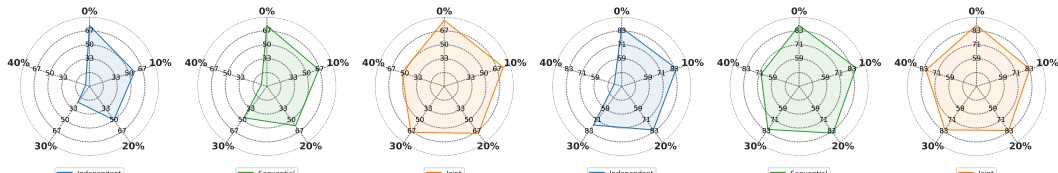

Figure 12: Task accuracy of CBMs trained with different strategies on the noisy CUB (left three) and AwA2 (right three) datasets. The radar charts illustrate task performance for the Independent (IND), Sequential (SEQ), and Joint (JOI) training paradigms under varying noise levels.

Table 5: Concept prediction accuracy of CBMs on noisy CUB and AwA2 datasets. Concept accuracies below 75% are **highlighted**, indicating significantly reduced interpretability under noise.

| | CUB | | | | | AwA2 | | | | |
|---|---|---|---|---|---|---|---|---|---|---|
| STRATEGY | $\gamma = 0.0$ | $\gamma = 0.1$ | $\gamma = 0.2$ | $\gamma = 0.3$ | $\gamma = 0.4$ | $\gamma = 0.0$ | $\gamma = 0.1$ | $\gamma = 0.2$ | $\gamma = 0.3$ | $\gamma = 0.4$ |
| IND | $96.5_{\pm0.0}$ | $93.8_{\pm0.0}$ | $91.6_{\pm0.0}$ | $89.1_{\pm0.1}$ | $85.4_{\pm0.2}$ | $78.5_{\pm0.8}$ | $78.4_{\pm0.8}$ | $78.1_{\pm0.7}$ | $77.3_{\pm0.3}$ | $75.3_{\pm0.8}$ |
| SEQ | $96.5_{\pm0.0}$ | $93.8_{\pm0.0}$ | $91.6_{\pm0.0}$ | $89.1_{\pm0.1}$ | $85.4_{\pm0.2}$ | $78.5_{\pm0.8}$ | $78.4_{\pm0.8}$ | $78.1_{\pm0.7}$ | $77.3_{\pm0.3}$ | $75.3_{\pm0.8}$ |
| JOI | $92.4_{\pm0.7}$ | $85.9_{\pm0.5}$ | $78.4_{\pm0.6}$ | $\mathbf{67.6_{\pm1.2}}$ | $\mathbf{57.3_{\pm0.3}}$ | $77.8_{\pm0.5}$ | $\mathbf{74.2_{\pm0.4}}$ | $\mathbf{70.1_{\pm0.8}}$ | $\mathbf{65.4_{\pm0.3}}$ | $\mathbf{57.4_{\pm0.2}}$ |

## C.5 Concept Selection Criteria

This section provides a detailed explanation of the concept selection criteria, extensively analyzed in previous work [43]. We present three criteria (Random, UCP, CCTP) addressed in the main paper, followed by additional criteria (LCP, ECTP, EUDTP) discussed in Appendix D.8.

i. **Random.** Concepts are selected uniformly at random as a baseline method, following Koh et al. [19]. Formally, each concept is assigned a random score: $s_i \sim \mathcal{U}(0, 1)$. This serves as a reference for evaluating other selection criteria.

ii. **Uncertainty of concept prediction (UCP).** Concepts with the highest prediction uncertainty are prioritized [21, 20]. Scores are assigned using entropy: $s_i = \mathcal{H}(\widehat{c}_i)$, where $\mathcal{H}$ denotes the entropy function. For binary concepts, this simplifies to $s_i = 1/|\widehat{c}_i - 0.5|$. This method selects concepts whose uncertainty potentially impairs target prediction accuracy.

iii. **Contribution on concept on target prediction (CCTP).** This criterion selects concepts based on their contributions to target predictions. Contributions are quantified as: $s_i = \sum_{j=1}^{M} |\widehat{c}_i \frac{\partial f_j}{\partial \widehat{c}_i}|$, where $f_j$ is the output for the $j$-th target class, and $M$ is the total number of classes. This approach is inspired by interpretability methods for neural networks [39].

iv. **Loss on concept prediction (LCP).** Concepts are selected based on the prediction loss compared to ground truth, defined as $s_i = |\widehat{c}_i - c_i|$. A lower prediction error typically correlates with improved target prediction. However, this method is impractical during inference since ground-truth labels are not available at test time.

v. **Expected change in target prediction (ECTP).** This criterion prioritizes concepts whose correction results in the greatest expected change in the target predictive distribution. Scores are defined as $s_i = (1 - \widehat{c}_i)D_{\text{KL}}(\widehat{y}_{\widehat{c}_i=0}\|\widehat{y}) + \widehat{c}_i D_{\text{KL}}(\widehat{y}_{\widehat{c}_i=1}\|\widehat{y})$ where $D_{\text{KL}}$ is the Kullback-Leibler divergence, and $\widehat{y}_{\widehat{c}_i=0}, \widehat{y}_{\widehat{c}_i=1}$ denote target predictions after interventions. The intuition is to intervene on concepts whose rectification significantly alters the target prediction [40].

vi. **Expected uncertainty decrease in target prediction (EUDTP).** Concepts that lead to the largest expected reduction in the entropy of the target predictive distribution upon intervention are selected. Formally, scores are defined as $s_i = (1 - \widehat{c}_i)\mathcal{H}(\widehat{y}_{\widehat{c}_i=0}) + \widehat{c}_i\mathcal{H}(\widehat{y}_{\widehat{c}_i=1}) - \mathcal{H}(\widehat{y})$. This method favors concepts that substantially reduce target prediction uncertainty when corrected [12].

# D  Additional Experiments

## D.1  Effect of Training Strategies under Noise

CBMs can be optimized with three training paradigms: *independent*, *sequential*, and *joint* (see Appendix C.2 for details). We evaluate their robustness by reporting task-level and concept-level accuracy across different noise ratios $\gamma$. Figure 12 presents task accuracy across training strategies,

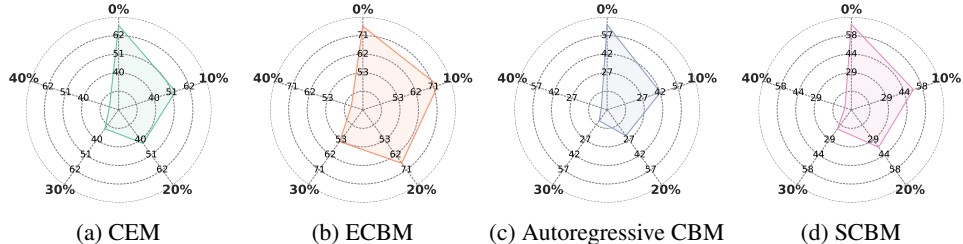

Figure 13: Task accuracy of CEM [61], ECBM [59], Autoregressive CBM [14], and SCBM [52] under varying noise levels on the CUB dataset. Task accuracy consistently drops as noise increases.

Table 6: Concept prediction accuracy of CBM variants on the noisy CUB dataset under BASE and SAM optimization. Concept prediction accuracy declines with increasing noise in all cases.

| METHOD | BASE | | | | | SAM | | | | |
|---|---|---|---|---|---|---|---|---|---|---|
| | $\gamma = 0.0$ | $\gamma = 0.1$ | $\gamma = 0.2$ | $\gamma = 0.3$ | $\gamma = 0.4$ | $\gamma = 0.0$ | $\gamma = 0.1$ | $\gamma = 0.2$ | $\gamma = 0.3$ | $\gamma = 0.4$ |
| CEM | 94.7±0.0 | 87.6±0.2 | 79.2±0.1 | 69.8±0.1 | 60.0±0.4 | 95.1±0.0 | 88.3±0.0 | 80.0±0.2 | 70.5±0.1 | 60.2±0.3 |
| ECBM | 96.4±0.2 | 95.7±0.2 | 93.5±0.3 | 90.2±0.4 | 85.7±0.5 | 96.9±0.1 | 95.5±0.5 | 94.1±0.0 | 90.8±0.2 | 86.4±0.2 |
| AR-CBM | 95.1±0.0 | 87.4±0.0 | 79.1±0.2 | 69.8±0.3 | 60.0±0.2 | 95.5±0.0 | 88.2±0.1 | 80.1±0.2 | 70.8±0.2 | 60.6±0.2 |
| SCBM | 94.6±0.0 | 89.7±0.0 | 83.2±0.0 | 73.6±0.3 | 62.1±0.2 | 94.7±0.1 | 90.2±0.0 | 84.2±0.1 | 75.7±2.4 | 66.5±6.7 |

with accompanying radar plots visualizing noise tolerance; larger and more regular polygons reflect greater robustness to noise.

Every training paradigm suffers noticeable degradation as label corruption intensifies. On AwA2, for instance, the joint model starts with the best clean accuracy but still drops from 88.8% to 81.7% when 40% of the labels are noisy. The independent and sequential variants fare even worse and largely collapse on CUB under the same noise level. Table 5 further reveals declines at the concept level. Although the joint paradigm better preserves task-level performance, its concept accuracy plunges below 75% once the noise ratio rises above roughly 30% on CUB and exceeds about 10% on AwA2. Because the joint objective simultaneously minimizes task and concept losses, it compromises concept fidelity as label quality declines. These findings corroborate earlier evidence of a performance–interpretability tradeoff [36] and underscore the need for training methods that preserve the robustness of CBM under noisy supervision.

## D.2 Robustness of CBM Variants under Noisy Supervision

We also investigate the effects of noise on several CBM variants, including CEM [61], ECBM [59], autoregressive CBM [14], and SCBM [52], trained on the CUB dataset with noise levels $\gamma \in [0.0, 0.4]$. Figure 13 presents task accuracy, while Table 6 reports concept prediction accuracy under both the standard Adam optimizer (BASE) and sharpness-aware minimization (SAM).

All CBM variants exhibit notable degradation in both task and concept accuracy as noise increases. Among them, ECBM achieves the highest concept accuracy across most settings, likely due to its energy-based formulation. However, its performance also deteriorates substantially under high noise levels, reaffirming the vulnerability of current CBM designs to noisy concept supervision. Training these variants with SAM provides modest but consistent improvements across all settings. These gains suggest that SAM offers a generalizable enhancement to CBM robustness. Nonetheless, the overall sensitivity of all variants highlights the ongoing need for more resilient training strategies. In this regard, SAM represents a promising direction for improving both performance and interpretability under noisy supervision.

## D.3 Additional Evidence of Disrupted Representation Alignment

In Section 3.2, we illustrated how noise disrupts the representation alignment for the class Le Conte Sparrow. Here, we provide additional qualitative results for the class Brewer Blackbird, which demonstrates a comparable degradation pattern.

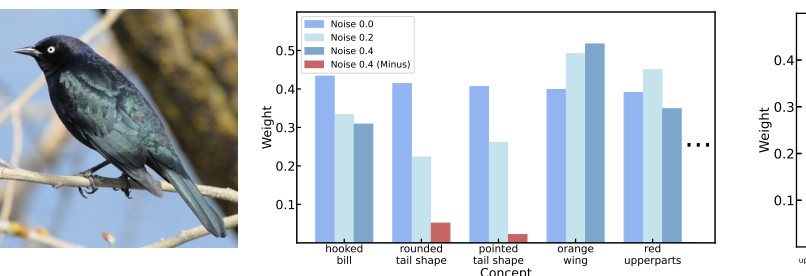

(a) Brewer Blackbird      (b) Importance shift in key concepts      (c) Key concepts under 40% noise

Figure 14: Effect of noise on representation alignment. (b) Top 5 most influential (*i.e.*, key) concepts for BREWER BLACKBIRD in a clean setting and tracks how their influence shifts as noise increases. (c) Top 5 influential concepts under 40% noise. In both figures, bars in blue denote positive weights, while red indicate negative weights, illustrating how noise reshapes concept importance.

As shown in Figure 14, at a moderate noise level (*i.e.*, 20%), the relative ordering of predictive dimensions remains quite well preserved. In contrast, at higher noise levels (*i.e.*, 40%), the saliency of informative dimensions deteriorates significantly, while irrelevant or spurious concepts become disproportionately influential. These findings further substantiate our conclusion that noise induces representational misalignment, and when this misalignment coincides with susceptible concepts, it becomes a key driver of instability in CBMs under noisy supervision.

## D.4 Robustness under Diverse Noise Scenarios

We further investigate CBM performance under realistic noise scenarios, specifically asymmetric [13] and grouped noise, using the CUB dataset. Asymmetric noise cyclically mislabels categories (*e.g.*, confusing similar animal species), while grouped noise randomly corrupts labels within similar concept groups (*e.g.*, wing color) as discussed in Appendix C.4.

As shown in Figure 15, CBMs experience substantial performance degradation under both noise types, broadly consistent with trends observed under symmetric noise. While both asymmetric and grouped noise lead to slightly more pronounced declines in concept accuracy

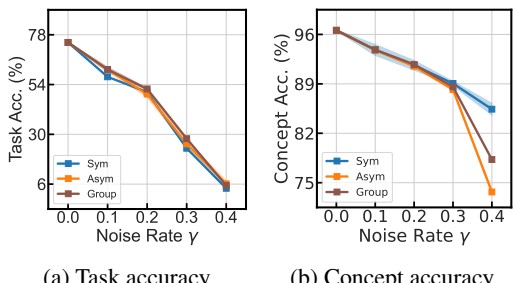

(a) Task accuracy      (b) Concept accuracy

Figure 15: Performance degradation under symmetric, asymmetric [13] and grouped noise on CUB dataset across different noise levels.

at higher corruption levels (*e.g.*, 40%), the overall performance patterns remain comparable. These findings indicate that CBMs are generally vulnerable to various forms of label noise, reinforcing the need for robustness strategies that extend beyond simple symmetric corruption and account for more realistic noise scenarios.

Furthermore, beyond introducing noise independently, we also considered more practical scenarios where concept and target noise can be correlated. To examine this case, we simulated a correlated noise model and evaluated the relationship between susceptibility and uncertainty. Specifically, we first corrupted the target labels, and concept noise was injected only when the target label was flipped, thereby inducing dependence between concept and target corruption. As illustrated in Figure 16, the positive relationship between uncertainty and susceptibility remains evident for susceptible concepts. This trend is consistent across different noise levels, suggesting that our analysis and mitigation approach remain effective under more realistic correlated noise conditions.

## D.5 Comparative Study of Noise Mitigation Methods

To validate the effectiveness of SAM in mitigating noise, we compare it with two representative baselines: label smoothing regularization (LSR) [47] and symmetric cross-entropy learning (SL) [56].

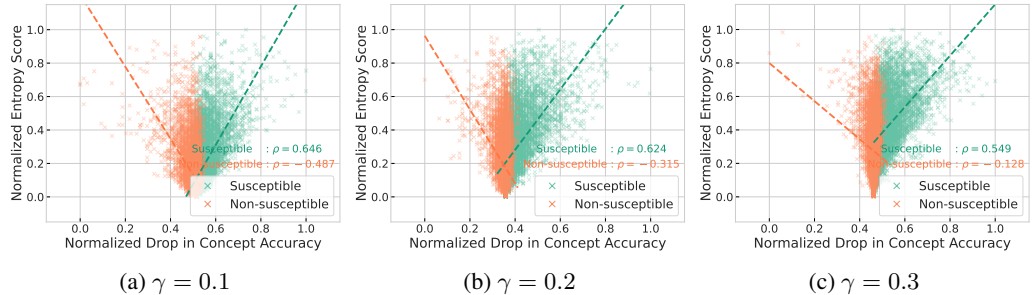

| (a) $\gamma = 0.1$ | (b) $\gamma = 0.2$ | (c) $\gamma = 0.3$ |

Figure 16: Correlation between predictive uncertainty and susceptibility on CUB under a realistic conditional noise model. The target label is corrupted first, and concept labels are corrupted only when the target has been perturbed. Pearson correlation coefficients ($\rho$) are reported separately for susceptible and non-susceptible sets, showing a positive correlation in the former.

LSR addresses overconfidence by softening class labels, while SL incorporates both standard and reverse cross-entropy losses to suppress the impact of noisy annotations.

Table 7 summarizes task and concept accuracy across noise levels. Across all conditions, SAM consistently outperforms LSR and SL. Under clean supervision, SAM achieves a task accuracy of 79.7%, surpassing LSR (72.8%) and SL (70.6%). Its advantages are more pronounced under noise: at 20% and 30% corruption, SAM outperforms the second-best method by 6.1% and 8.4% in concept accuracy, respectively.

Table 7: Comparison of concept and task accuracies across diverse label-noise mitigation methods on CUB.

| METHOD | METRIC | NOISE RATIO | | | |
|---|---|---|---|---|---|
| | | $\gamma = 0.0$ | $\gamma = 0.1$ | $\gamma = 0.2$ | $\gamma = 0.3$ |
| LSR | CONCEPT ACC | 95.9±0.1 | 91.3±0.0 | 86.5±0.2 | 70.7±7.7 |
| | TASK ACC | 72.8±0.4 | 60.4±0.4 | 46.8±1.6 | 15.0±12.9 |
| SL | CONCEPT ACC | 95.0±0.1 | 91.5±0.1 | 86.6±0.2 | 81.6±0.2 |
| | TASK ACC | 70.6±1.3 | 60.5±0.4 | 48.0±1.7 | 27.0±7.5 |
| SAM | CONCEPT ACC | 97.3±0.0 | 95.1±0.1 | 92.7±0.1 | 90.0±0.1 |
| | TASK ACC | 79.7±0.1 | 67.7±0.6 | 50.4±2.0 | 26.6±1.2 |

These results demonstrate the robustness of SAM in noisy settings. While our evaluation focuses on selected methods, other techniques (*e.g.*, co-teaching [48] or MentorNet [17]) offer promising alternatives through sample selection and teacher-student training. Integrating such techniques with CBMs represents a valuable direction for future work. Overall, these results position SAM as an effective and scalable approach for robust concept-based learning, as further supported by our theoretical insights in Appendix A.

### D.6 Evaluating Concept Noise in LLM-Generated Annotations

Recent studies have explored using large language models (LLMs) to automatically generate concept sets for CBMs [60, 27, 45]. We identify two main approaches to assess the noisiness of such annotations: (i) evaluating whether LLMs can define concepts accurately by comparing them to expert-curated datasets, and (ii) assess-

Table 8: Performance comparison between LLM-generated and ground-truth concept annotations.

| CONCEPT | ANNOTATION SIM. | TASK ACC. |
|---|---|---|
| LLM-BASED | 21.8 | 68.5 |
| GROUND-TRUTH | - | 74.3 |

ing whether LLMs can reliably annotate individual samples using a fixed expert-defined concept set. Here, we focus on the second approach. Using the concept vocabulary from the CUB dataset, we prompt an LLM (*i.e.*, ChatGPT [28]) to annotate each target class.

As shown in Table 8, the resulting annotations agree with human-labeled ground truth only 21.8% of the time, revealing a significant misalignment. This highlights a key issue, *i.e.*, general-purpose LLMs often produce annotations that diverge from expert standards, introducing a new source of noise. To assess the effect of this noise, we trained a CBM on the LLM-annotated dataset. This model achieved 68.5% accuracy, a 5.8% drop from the baseline trained on expert annotations. Unlike traditional concept noise, which often stems from human inconsistency, LLM-induced noise may arise from the the limited domain grounding and generalization of model. These findings expose the current limitations of LLM-generated concept supervision and reinforce the broader challenge of label noise in building robust and interpretable CBMs, particularly in high-stakes domains.

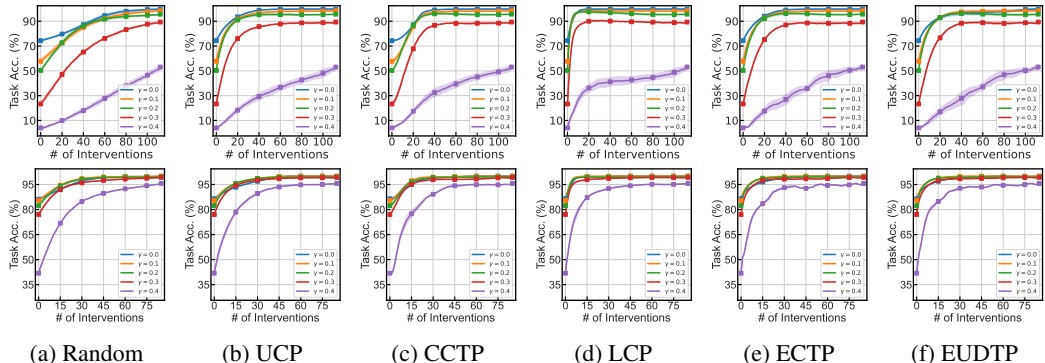

Figure 17: Impact of noise on the intervention effectiveness of CBMs using Random, UCP, LCP, CCTP, ECTP, EUDTP strategies under varying noise levels. The top row represents the CUB dataset, and the bottom row shows results for AwA2 dataset.

## D.7 Effect of Target Predictor Complexity

Here, we examine how the structure of the target predictor $f$ affects robustness under noisy conditions. A central premise in CBMs is that a linear predictor promotes interpretability; however, this simplicity may come at the cost of reduced expressiveness, particularly in tasks requiring more complex decision boundaries. This trade-off makes the choice of $f$ a critical design factor. To explore this, we replace the standard linear predictor with a shallow non-linear alternative (*i.e.*, a two-layer feedforward network) and train both models under identical conditions.

Table 9: Performance comparison of the $f$ model under different noise types and levels of non-linearity.

| | | NOISE | | |
|---|---|---|---|---|
| OPTIMIZER | LINEARITY | $\gamma = 0.0$ | $\gamma = 0.2$ | $\gamma = 0.4$ |
| BASE | LINEAR | 74.3±0.3 | 50.3±0.7 | 4.0±0.7 |
| | NON-LINEAR | 73.5±0.2 | 47.4±2.1 | 3.6±0.3 |
| SAM | LINEAR | 79.0±0.8 | 54.2±0.7 | 5.0±1.4 |
| | NON-LINEAR | 78.3±0.5 | 51.6±1.8 | 4.1±0.3 |

As shown in Table 9, the non-linear variant consistently underperforms its linear counterpart, with performance degrading more severely as noise increases. These results suggest that non-linear predictors are more susceptible to overfitting noisy concept representations, thereby undermining generalization. Due to the concept bottleneck imposed by CBMs, increasing the complexity of $f$ does not inherently enhance predictive accuracy; rather, it can exacerbate the propagation of noise from the concept layer to the output. Overall, these findings indicate that under noisy conditions, simpler target predictors may offer greater robustness. However, this conclusion is drawn from a preliminary comparison between a linear and a two-layer model. Exploring a broader range of non-linear architectures and incorporating regularization techniques may offer valuable insights into the trade-off between expressiveness and robustness.

## D.8 Impact of Noise on Intervention Effectiveness

As outlined in Appendix C.5, we evaluate six concept selection strategies for intervention: Random, UCP, CCTP, LCP, ECTP, and EUDTP. While the main paper focuses on the first three, here we provide an extended analysis of the remaining strategies (*e.g.*, LCP, ECTP, and EUDTP) for a more comprehensive evaluation.

Figure 17 presents task accuracy improvements achieved through concept-level interventions on the CUB and AwA2 datasets under varying levels of noise. Across both datasets, we observe a consistent pattern: the effectiveness of interventions diminishes as the noise level increases, irrespective of the intervention selection strategy. These findings corroborate our main results and further demonstrate that noise significantly compromises the reliability of concept-based interventions. Moreover, for all intervention methods, the upper bound on task accuracy (which corresponds to interventions applied to all concepts) systematically decreases with increasing noise levels. This downward trend underscores the importance of incorporating robustness-aware selection criteria when designing intervention strategies for noisy environments.

Table 10: Comparison of concept and task accuracy between BASE and SAM on the CUB and AwA2.

| TRAINING STRATEGY | OPTIMIZER | METRIC | NOISE | | | | Δ |
| | | | $\gamma = 0.1$ | $\gamma = 0.2$ | $\gamma = 0.3$ | $\gamma = 0.4$ | |
|---|---|---|---|---|---|---|---|
| | | | CUB [54] DATASET | | | | |
| INDEPENDENT | BASE | CONCEPT ACC | $93.8_{\pm0.1}$ | $91.6_{\pm0.0}$ | $89.1_{\pm0.0}$ | $85.4_{\pm0.0}$ | |
| | | TASK ACC | $57.7_{\pm2.0}$ | $50.3_{\pm0.7}$ | $23.3_{\pm1.2}$ | $4.0_{\pm0.7}$ | |
| | SAM | CONCEPT ACC | $94.6_{\pm0.1}$ | $92.5_{\pm0.1}$ | $89.7_{\pm0.1}$ | $86.3_{\pm0.1}$ | **+0.8** |
| | | TASK ACC | $61.8_{\pm1.8}$ | $54.2_{\pm0.7}$ | $28.5_{\pm1.4}$ | $5.0_{\pm1.4}$ | **+3.6** |
| SEQUENTIAL | BASE | CONCEPT ACC | $93.8_{\pm0.0}$ | $91.6_{\pm0.0}$ | $89.1_{\pm0.0}$ | $85.4_{\pm0.1}$ | |
| | | TASK ACC | $66.6_{\pm0.4}$ | $59.3_{\pm0.6}$ | $47.0_{\pm1.7}$ | $6.1_{\pm2.6}$ | |
| | SAM | CONCEPT ACC | $94.6_{\pm0.1}$ | $92.5_{\pm0.1}$ | $89.7_{\pm0.1}$ | $86.3_{\pm0.1}$ | **+0.8** |
| | | TASK ACC | $70.5_{\pm0.6}$ | $63.5_{\pm0.9}$ | $50.1_{\pm1.1}$ | $10.7_{\pm6.0}$ | **+4.0** |
| JOINT | BASE | CONCEPT ACC | $85.9_{\pm0.5}$ | $78.4_{\pm0.6}$ | $67.6_{\pm1.2}$ | $57.3_{\pm0.3}$ | |
| | | TASK ACC | $75.2_{\pm0.3}$ | $69.2_{\pm0.5}$ | $59.8_{\pm0.3}$ | $50.1_{\pm0.5}$ | |
| | SAM | CONCEPT ACC | $86.0_{\pm0.2}$ | $78.5_{\pm0.1}$ | $68.0_{\pm0.8}$ | $57.9_{\pm0.3}$ | **+0.3** |
| | | TASK ACC | $76.1_{\pm0.4}$ | $69.9_{\pm0.6}$ | $60.8_{\pm0.4}$ | $50.6_{\pm1.5}$ | **+0.8** |
| | | | AwA2 [58] DATASET | | | | |
| INDEPENDENT | BASE | CONCEPT ACC | $78.4_{\pm0.8}$ | $78.1_{\pm0.7}$ | $77.3_{\pm0.3}$ | $75.3_{\pm0.8}$ | |
| | | TASK ACC | $85.5_{\pm0.3}$ | $82.3_{\pm1.4}$ | $77.1_{\pm0.5}$ | $41.9_{\pm1.0}$ | |
| | SAM | CONCEPT ACC | $78.6_{\pm0.7}$ | $78.5_{\pm0.5}$ | $77.9_{\pm0.7}$ | $75.9_{\pm1.3}$ | **+0.5** |
| | | TASK ACC | $88.1_{\pm0.1}$ | $85.7_{\pm0.4}$ | $78.6_{\pm2.8}$ | $46.5_{\pm1.4}$ | **+3.0** |
| SEQUENTIAL | BASE | CONCEPT ACC | $78.4_{\pm0.8}$ | $78.1_{\pm0.7}$ | $77.3_{\pm0.3}$ | $75.3_{\pm0.8}$ | |
| | | TASK ACC | $87.6_{\pm0.3}$ | $85.8_{\pm0.3}$ | $81.8_{\pm1.1}$ | $70.1_{\pm3.9}$ | |
| | SAM | CONCEPT ACC | $78.6_{\pm0.7}$ | $78.5_{\pm0.5}$ | $77.9_{\pm0.7}$ | $75.9_{\pm1.3}$ | **+0.5** |
| | | TASK ACC | $89.5_{\pm0.5}$ | $88.0_{\pm0.5}$ | $82.6_{\pm3.1}$ | $69.6_{\pm6.3}$ | **+1.1** |
| JOINT | BASE | CONCEPT ACC | $74.2_{\pm0.4}$ | $70.1_{\pm0.8}$ | $65.4_{\pm0.3}$ | $57.4_{\pm0.2}$ | |
| | | TASK ACC | $84.2_{\pm0.1}$ | $83.0_{\pm0.3}$ | $82.2_{\pm0.1}$ | $81.7_{\pm0.3}$ | |
| | SAM | CONCEPT ACC | $74.9_{\pm0.4}$ | $72.7_{\pm0.6}$ | $67.7_{\pm0.7}$ | $58.9_{\pm0.9}$ | **+1.8** |
| | | TASK ACC | $89.0_{\pm0.1}$ | $88.4_{\pm0.2}$ | $87.3_{\pm0.2}$ | $86.6_{\pm0.3}$ | **+5.1** |

## D.9 Comparison of Prediction Accuracy between SAM and BASE Optimizer

Table 10 reports the overall task and concept prediction accuracy of CBMs trained with SAM and the baseline (BASE) on the CUB and AwA2 datasets under various noise settings ($\gamma \in \{0.0, 0.2, 0.4\}$). Across all noise levels and types, SAM consistently outperforms BASE in both concept and task accuracy, demonstrating its robustness to noise.

These results suggest that SAM is more effective than BASE at mitigating the impact of noisy labels, maintaining higher accuracy across different CBM training paradigms. On the CUB dataset, for example, SAM yields notable gains for both independent and sequential models, achieving average improvements of $0.8\%$ and $3.6\%$ in concept and task accuracy for independent, and $0.8\%$ and $4.0\%$ for sequential, respectively. On AwA2, the largest improvements are observed in the joint setting, with increases of $1.8\%$ in concept accuracy and $5.1\%$ in task accuracy. Overall, these findings confirm that SAM enhances CBM robustness across datasets and training strategies, effectively reducing the adverse effects of noise irrespective of dataset characteristics.

## D.10 Robustness of CBMs across Backbone Architectures

To assess the impact of backbone architecture on the robustness of CBMs under noise, we conduct experiments using two distinct architectures: ResNet-18 [15] and ViT-B/16 [9]. These models are trained on the CUB dataset with varying levels of noise ($\gamma \in \{0.1, 0.2, 0.3, 0.4\}$). We also compare the performance of the standard SGD (i.e., BASE) optimizer with the SAM optimizer [10].

Table 11 presents the task and concept prediction accuracies for each configuration. Our findings indicate that, regardless of the backbone architecture, CBMs experience significant performance degradation as noise increases. However, models trained with the SAM optimizer consistently exhibit improved robustness compared to those trained with SGD. Specifically, SAM yields average task accuracy improvements of $5.07\%$ for ResNet-18 and $7.31\%$ for ViT-B/16 under noisy conditions.

Table 11: Impact of noise on different CBM backbones. CBMs implemented with ResNet-18 and ViT-B/16 exhibit notable performance degradation under increasing levels of noise. The application of SAM consistently alleviates this degradation across all noise settings.

| | | BASE | | | SAM | | | |
|---|---|---|---|---|---|---|---|---|
| BACKBONE | METRIC | $\gamma = 0.0$ | $\gamma = 0.2$ | $\gamma = 0.4$ | $\gamma = 0.0$ | $\gamma = 0.2$ | $\gamma = 0.4$ | $\Delta$ |
| RESNET-18 | CONCEPT ACC | 95.23 | 90.40 | 81.28 | 95.98 | 92.70 | 81.78 | **+1.19** |
| | TASK ACC | 69.14 | 49.14 | 0.90 | 73.32 | 60.55 | 0.52 | **+5.07** |
| VIT-B/16 | CONCEPT ACC | 96.04 | 89.06 | 82.76 | 96.74 | 90.95 | 85.84 | **+1.89** |
| | TASK ACC | 73.66 | 31.05 | 1.69 | 77.87 | 47.26 | 3.19 | **+7.31** |

These results align with the observations of Chen et al. [4], who demonstrated that ViTs are prone to converging to sharp local minima, making them more sensitive to optimization challenges. The application of SAM, which promotes flatter loss landscapes, effectively mitigates this issue, leading to enhanced generalization and robustness. Consequently, while the choice of backbone architecture influences the degree of sensitivity to noise, the adoption of SAM emerges as a critical factor in enhancing the resilience of CBMs to noisy annotations.

### D.11 Additional Evidence that Susceptibility Approximates Uncertainty

In Section 4.2, we demonstrated on the CUB dataset that susceptibility can be effectively approximated by uncertainty. Here, we provide additional results on the AwA2 dataset, which exhibit similar trends.

As shown in Figure 18, a positive correlation between uncertainty and susceptibility consistently appears across datasets. While the AwA2 dataset shows a mild negative correlation within the non-susceptible set, this does not contradict the strong positive relationship observed among susceptible concepts. These findings support that our analysis captures a general and dataset-independent pattern.

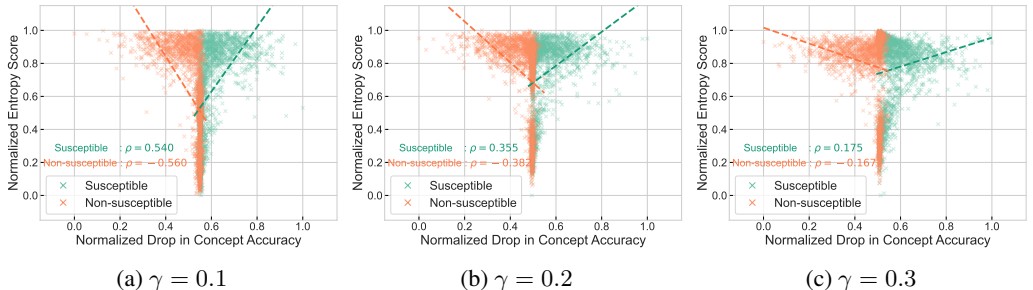

(a) $\gamma = 0.1$         (b) $\gamma = 0.2$         (c) $\gamma = 0.3$

Figure 18: Correlation between uncertainty and susceptibility on AwA2. As with the CUB dataset, Pearson correlation coefficients ($\rho$) are reported separately for susceptible and non-susceptible sets, showing a positive correlation in the former. A negative correlation appears in the non-susceptible subset but does not alter the positive relationship observed for the susceptible subset.

### D.12 Concept-Level Qualitative Results

In Section 2.3, we presented qualitative t-SNE [51] visualizations to illustrate how noise deteriorates the interpretability of learned concept representations. Here, we provide additional qualitative results focusing on a subset of representative concepts related to bird morphology and coloration. Specifically, we examine the spatial clustering patterns of concept predictions for various bill shapes (*e.g.*, `Curved Bill`, `Spatulate Bill`, and `All Purpose Bill`) as well as wing colors (*e.g.*, `Blue Wings` and `Orange Wings`). These concepts were selected based on their semantic diversity and prevalence in the dataset. The resulting visualizations further confirm that increased noise disrupts the cluster structure of concept embeddings, reducing the ability of model to maintain separable and interpretable representations.

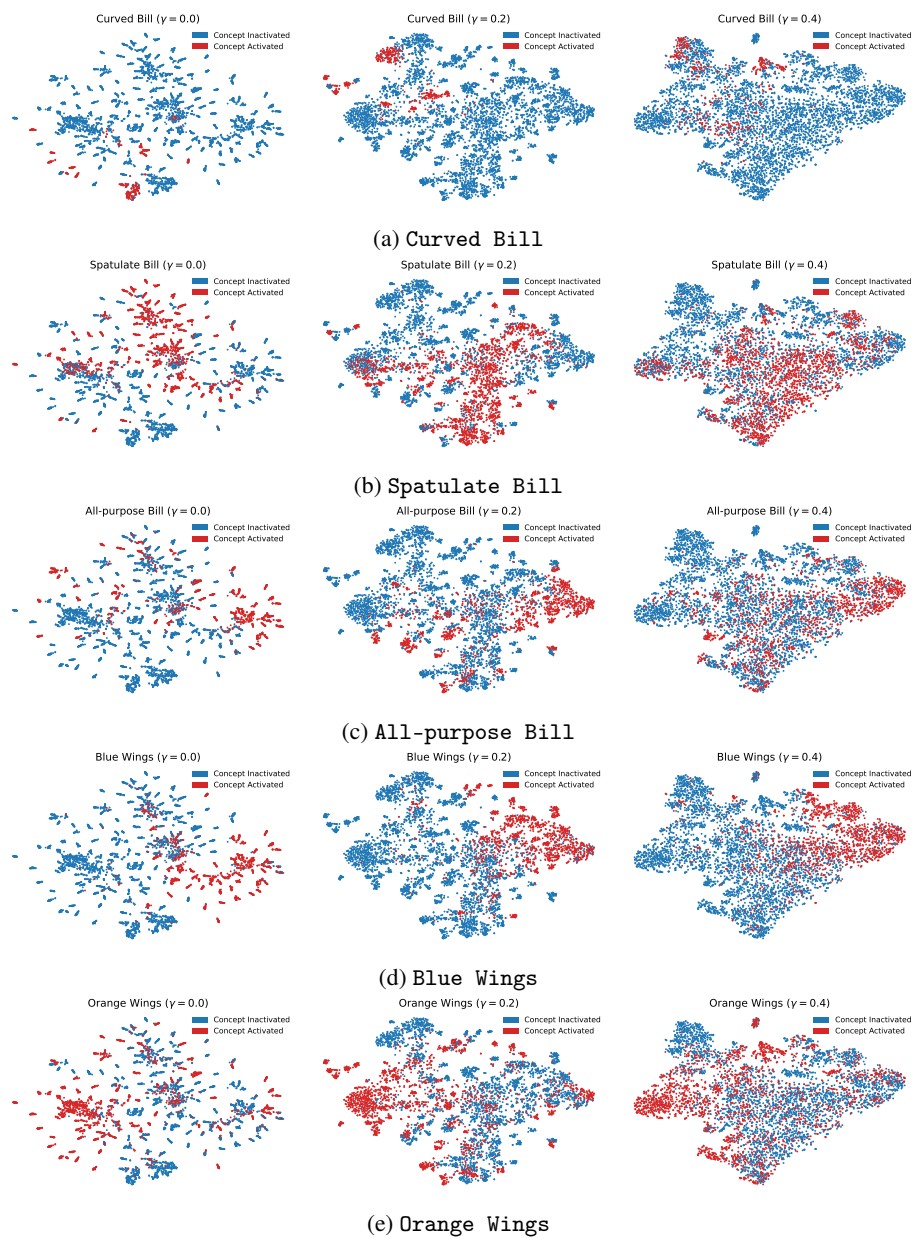

Figure 19: Qualitative results: t-SNE [51] visualizations of various concept embeddings learnt in CUB with sample points colored red if the concept is active and blue if the concept is inactive in that sample. As noise ratio increases, the concepts are clearly entangled making concepts unreliable.

## D.13 Noise Injection Protocol

In Figure 6, noise is injected at the concept level with different random seeds, which can substantially influence how the results are interpreted. To assess whether the same concepts are consistently identified as susceptible across different noise realizations (or whether susceptibility varies randomly) we report the susceptible concepts obtained for each seed in Figure 20. We find that similar concepts are repeatedly identified as susceptible across seeds, indicating structural sensitivity rather than random variation.

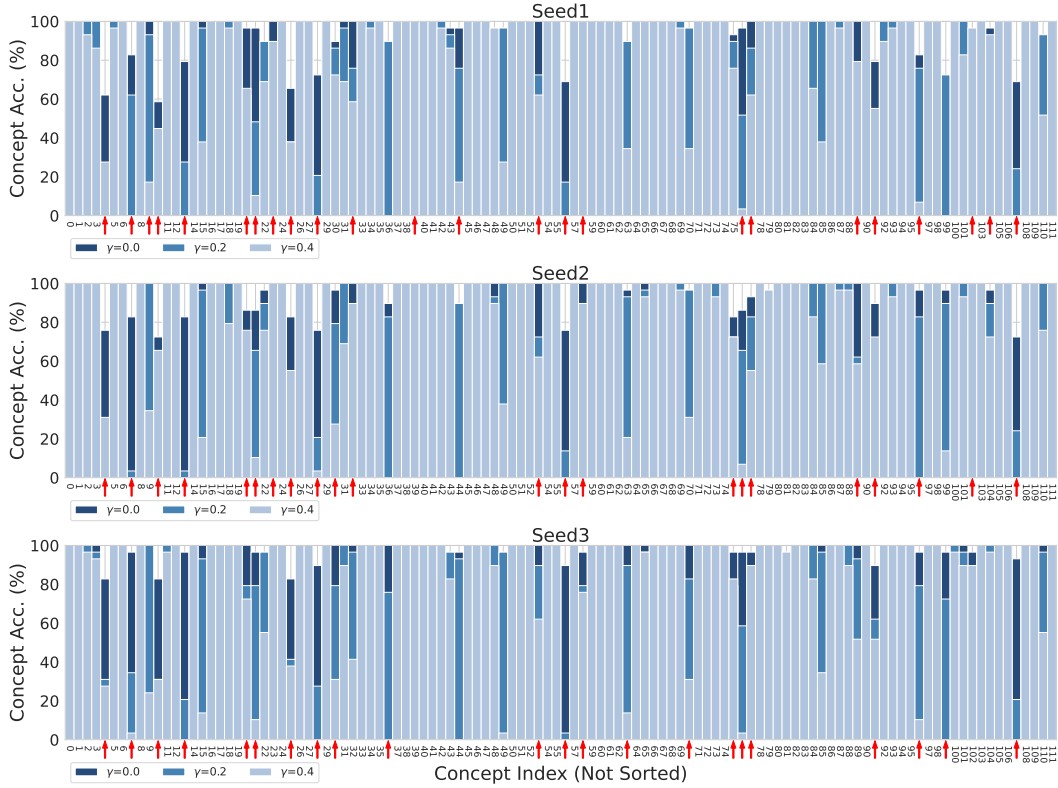

Figure 20: Susceptible concept sets across random seeds. Susceptible concept sets are identified for each random seed, and similar concepts are consistently detected across seeds.

