# OpenReview forum: "An Analysis of Concept Bottleneck Models: Measuring, Understanding, and Mitigating the Impact of Noisy Annotations"
_NeurIPS.cc/2025/Conference — NeurIPS 2025 poster_

### Official Review · Reviewer_AKL9 · 2025-06-05

**Clarity:** 3
**Significance:** 3
**Originality:** 3
**Rating:** 5
**Confidence:** 4

**Summary:**

This paper investigates the impact of concept-level noise in Concept Bottleneck Models (CBMs), focusing on a realistic scenario where the supervision signal on intermediate concepts is imperfect. The authors conduct a systematic study on how concept noise affects downstream prediction performance, interpretability, and intervention effectiveness. To mitigate the model’s vulnerability to noisy concepts, they propose two complementary techniques: sharpness-aware minimization during training, and uncertainty-based intervention during inference. Experimental results across multiple datasets demonstrate the effectiveness of the proposed solutions.

**Questions:**

The noise simulation process appears oversimplified. As described in the paper: "each binary concept label $c_i \in {0,1}$ is independently flipped with probability $\gamma$ to simulate concept noise $\hat{c}$. For target noise $\hat{y}$, each class label $y$ is randomly changed to a different label, uniformly sampled from the remaining classes, also with probability $\gamma$."

However, as the authors themselves note in lines 31–33, "changing concepts can shift the label." This indicates that concept and target label noise are not necessarily independent in real-world scenarios. Given this, how do the authors justify the assumption of independence between noisy concepts and noisy labels in their experimental setup? Would the proposed methods remain effective under a more realistic, correlated noise model?

**Ethical Concerns:**

["NO or VERY MINOR ethics concerns only"]

**Final Justification:**

It mainly addresses my previous concerns. So I will raise my original score from boardline accept to accept.

**Limitations:**

yes

**Quality:**

3

**Strengths And Weaknesses:**

Strenghts:

1. The paper offers a systematic and well-structured investigation into how concept noise influences various aspects of CBMs.

2. The writing is clear and logically organized, making the paper easy to follow.

3. The proposed methods are simple, practical, and effective in improving model robustness under noise.

Weaknesses:

1. The assumptions regarding the noise generation process are arguably too simplistic. In Section 2.1, this process ignores potential dependencies between concepts and labels.

---

> ### Author Rebuttal · Authors · 2025-07-31
>
> We sincerely thank the reviewer for acknowledging our works as practical and well-organized, and for providing thoughtful feedback. Below, we address the specific comments and welcome any further discussion.
>
> ---
>
> **On the independence assumption**
> > The assumptions regarding the noise generation process are arguably too simplistic. In Section 2.1, this process ignores potential dependencies between concepts and labels.
> > The noise simulation process appears oversimplified. As described in the paper: “each binary concept label $c_i \in \{0, 1\}$ is independently flipped with probability $\gamma$ to simulate concept noise $\hat{c}$. For target noise $\hat{y}$, each class label $y$ is randomly changed to a different label, uniformly sampled from the remaining classes, also with probability $\gamma$.
> > However, as the authors themselves note in lines 31-33, “changing concepts can shift the label.” This indicates that concept and target label noise are not necessarily independent in real-world scenarios. Given this, how do the authors justify the assumption of independence between noisy concepts and noisy labels in their experimental setup? Would the proposed methods remain effective under a more realistic, correlated noise model?
>
> Thank you for this insightful comment. Our primary objective is to provide a systematic analysis of how noise affects the performance and interpretability of CBMs. To isolate basic effects and to evaluate mitigation strategies in a controlled setting, we sought an experimental design in which concept noise and target noise can be manipulated independently. This design allows us to vary each factor without confounding and to attribute observed changes to a single source of corruption. For this reason, we considered the independence assumption a reasonable and transparent starting point for an initial analysis across a range of noise levels.
>
> We agree that in many practical settings concept and target noise can be correlated. To examine this case, we additionally simulated a correlated noise model and evaluated susceptibility and its connection to uncertainty. Concretely, we first corrupted the target label; only when the target changed did we inject concept noise, thereby inducing dependence between concept and target corruption. The results are summarized below.
>
> `Correlation between uncertainty and susceptibility on practical noise`
> | Noise Ratio | Susceptible Concepts | Non-susceptible Concepts |
> |---|---|---|
> | $\gamma$ = 0.1 | 0.646 | -0.487 |
> | $\gamma$ = 0.2 | 0.624 | -0.315 |
> | $\gamma$ = 0.3 | 0.549 | -0.128 |
>
>
> Even under dependent noise, the positive association between uncertainty and susceptibility persists for susceptible concepts. This pattern is consistent across noise levels and indicates that our analysis and mitigation approach remain effective in a more realistic correlated noise setting.

---

> > ### Comment · Reviewer_AKL9 · 2025-08-05
> >
> > Thank you for the author's rebuttal. It mainly addresses my previous concerns. I strongly encourage the authors to include these additional results in the final manuscript and to revise the introduction accordingly so that the paper presents a more complete narrative. I will therefore raise my original score from boardline accept to accept.

---

> > > ### Author Response · Authors · 2025-08-06
> > >
> > > We are pleased that our response has adequately addressed the reviewer’s earlier concerns. In the final version, we will provide a clearer explanation and additional experiments regarding `On the independence assumption`, as suggested. We appreciate this constructive feedback and are grateful for the improved score, which strengthens our chances of acceptance.

---

### Official Review · Reviewer_WqB3 · 2025-06-29

**Clarity:** 3
**Significance:** 2
**Originality:** 1
**Rating:** 5
**Confidence:** 3

**Summary:**

This paper provides an empirical analysis, backed up by some theoretical arguments, of how concept bottleneck models (CBMs) are affected by noisy concept annotations. The authors identify noisy concept supervision in training data as a major source of downstream performance degradation. They claim that this effect is primarily driven by a subset of so-called “susceptible concepts.” To mitigate the issue, the authors propose two strategies: sharpness-aware minimization (SAM) during training and uncertainty-guided interventions at inference. Experiments on two real-world datasets support their conclusions.

**Questions:**

1. The theory in Appendix B compares $f(\hat{c}_i)$ with $f(c^*_i)$, focusing on the error in predicting concept $i$. However, how does this prediction error relate to the *training noise rate*, which is the central focus of this paper? The setup seems to conflate inference-time uncertainty with training-time corruption—could the authors clarify this connection?
2. Related to W1), how do the authors position their findings in light of prior work (e.g., Shin et al., Penaloza et al., Sheth & Kahou), which also studied intervention strategies and robustness in CBMs? What distinguishes this work?
3. The definition of susceptible concepts is based on a greater-than-average drop in accuracy under noise. But what if a concept suffers a large decline yet contributes negligibly to the final prediction (e.g., in a linear bottleneck model, if its weight is close to zero)? Shouldn't the uncertainty-aware intervention take into account concept relevance?

**Ethical Concerns:**

["NO or VERY MINOR ethics concerns only"]

**Final Justification:**

Author's responses and additional experiments addressed my concerns.

**Limitations:**

The paper lists several limitations which are fair.

---

Overall, this is a well-written and timely paper that provides a useful empirical analysis of CBMs under noisy supervision. However, in my opinion, it falls short of the NeurIPS bar in terms of novelty and conceptual contribution. The main techniques used are existing ones, and the findings, while insightful, are largely expected. I would encourage the authors to consider submitting to a workshop or resubmission after strengthening the theoretical and/or methodological contributions.

**Paper Formatting Concerns:**

No concerns

**Quality:**

3

**Strengths And Weaknesses:**

### Strengths

1. The paper presents a systematic empirical study of the effects of noise in CBMs, including detailed analyses of prediction performance, interpretability, and intervention effectiveness.
2. It offers practical insights for working with noisy concept annotations.
3. The study is timely, especially given the growing use of LLM-generated concept labels, which are prone to noise.
4. The theoretical derivations connecting susceptibility and predictive uncertainty provide additional support and motivation for the intervention strategy.

### Weaknesses

1. **Lack of Novel Methodology.** The paper does not introduce any fundamentally new methods. Both SAM and uncertainty-based interventions are existing techniques that are simply applied to this setting. The novelty is limited to empirical observations and analysis, which—though valuable—do not represent a significant conceptual advance.
2. **Limited number of datasets.** For a paper whose main contributions are empirical, the use of only two datasets (CUB and AwA2) is limiting. The findings would be more convincing if demonstrated on a broader range of settings.
3. **Predictable Findings.** Several results—such as the degradation of interpretability and performance under noise, or the benefits of targeting uncertain predictions—are somewhat expected. While clearly and rigorously shown, these insights are incremental.

---

> ### Author Rebuttal · Authors · 2025-07-31
>
> We sincerely thank the reviewer for recognizing the practicality and timeliness of our work and providing thoughtful feedback. Below, we address the specific comments and welcome any further discussion.
>
> ---
>
> **Significance of our work**
> > The paper does not introduce any fundamentally new methods. Both SAM and uncertainty-based interventions are existing techniques that are simply applied to this setting. The novelty is limited to empirical observations and analysis, which–though valuable–do not represent a significant conceptual advance.
>
> We appreciate the constructive comment. The significance of our study can be summarized in three parts.
>
> * To our knowledge, this is the first systematic analysis of the noisy concept annotation problem in CBMs, a practical issue that can arise from human error and subjective judgment yet has remained largely unattended. We show, through multi-faceted analyses, that such noise negatively affects all core CBM properties: interpretability, intervention effectiveness, and predictive performance.
>
> * We find that degradation is not uniform across concepts. Rather, it concentrates on a subset that is **susceptible** to noise. This small set explains most of the accuracy decline and therefore drives the overall performance drop.
>
> * To mitigate the risks posed by susceptible concepts, we propose to use sharpness-aware minimization during training and uncertainty-guided intervention at inference. We provide broad empirical results and theoretical support demonstrating that these strategies are effective at restoring the reliability of CBMs.
>
> ---
>
> **Unexpected findings**
> > Several results–such as the degradation of interpretability and performance under noise, or the benefits of targeting uncertain predictions–are somewhat expected. While clearly and rigorously shown, these insights are incremental.
>
> We appreciate the reviewer’s point. This work, however, presents results that were not previously documented and are difficult to anticipate.
>
> First, it is commonly believed that CBMs allow one to correct the final prediction via intervention even when a concept is mispredicted. From this view, one might expect that models trained with noisy concepts would readily recover performance through intervention. Our experiments contradict this expectation. As shown in Section 2.4, when CBMs are trained on datasets with concept noise, the intervention effectiveness itself is substantially reduced, so the expected recovery does not materialize. Given that real deployments often rely on concept sources with nontrivial noise, this indicates that intervention may fail to operate as intended and can lead to unexpected failure.
>
> Second, CBMs degrade in a qualitatively different manner from standard DNNs under the same noise level. For example, on the CUB classification task with 0%, 20%, and 40% noise, a DNN exhibits a relatively moderate decline (79.7 → 65.9 → 53.0), whereas a CBM shows an abrupt collapse (74.3 → 50.3 → 4.0). This occurs because the two-stage structure of CBM propagates errors in concept prediction directly to the target predictor. The evidence shows that CBMs are considerably more vulnerable to concept noise than is typically assumed.
>
> These findings reveal an inherent sensitivity of CBMs to concept noise and underscore the need to recognize this vulnerability and develop appropriate mitigation in order to build reliable CBMs.
>
> ---
>
> **Positioning relative to prior work**
> > how do the authors position their findings in light of prior work (e.g., Shin et al., Penaloza et al., Sheth & Kahou), which also studied intervention strategies and robustness in CBMs? What distinguishes this work?
>
> We appreciate the question.
>
> **Shin et al. [1]**: The focus is different from ours. They propose and evaluate various intervention strategies and analyze their effects. Our work systematically analyzes how noise degrades CBM behavior and proposes practical mitigation for this noise setting. We further show, both theoretically and empirically, that uncertainty-guided intervention approximates a susceptibility-based intervention, supporting its utility under noisy supervision.
>
> **Penaloza et al. [2]**: They share the motivation that CBMs can be sensitive to label noise and propose model improvements, **assuming they are vulnerable to noise**. In contrast, our contribution is a comprehensive analysis of _how_ concept noise affects interpretability, intervention effectiveness, and predictive performance, together with an integrated mitigation framework that operates at both training and inference.
>
> **Sheth and Ebrahimi Kahou [3]**: Their goal is different. They investigate that CBMs can learn irrelevant concept representations and target robustness under distribution shift. We study concept noise that arises naturally in practice, identify the key drivers of failure through a systematic analysis, and propose simple and effective strategies to mitigate the resulting risks.
>
> ---
>
> **Broader range of settings**
> > For a paper whose main contributions are empirical, the use of only two datasets (CUB and AwA2) is limiting. The findings would be more convincing if demonstrated on a broader range of settings.
>
> We appreciate this concern and will include additional results in the final version. We conducted two complementary experiments in the tables below.
>
> Although the main paper analyzes susceptibility primarily on CUB, we replicated the analysis on AwA2. We observe that the positive correlation between uncertainty and susceptibility persists. While the non-susceptible set shows a more negative correlation in AwA2, this does not weaken the positive correlation within the susceptible set.
>
> To better reflect practical conditions where concept and target may be dependent, we designed a dependent noise model. We first corrupt target labels, then add concept noise only when the target changes. The positive correlation between susceptibility and uncertainty remains under this more realistic setting.
>
> `Correlation between uncertainty and susceptibility on AwA2`
> | Noise Ratio | Susceptible Concepts | Non-susceptible Concepts |
> |---|---|---|
> | $\gamma$ = 0.1 | 0.540 | -0.560 |
> | $\gamma$ = 0.2 | 0.355 | -0.382 |
> | $\gamma$ = 0.3 | 0.175 | -0.167 |
>
>
> `Correlation between uncertainty and susceptibility on practical noise`
> | Noise Ratio | Susceptible Concepts | Non-susceptible Concepts |
> |---|---|---|
> | $\gamma$ = 0.1 | 0.646 | -0.487 |
> | $\gamma$ = 0.2 | 0.624 | -0.315 |
> | $\gamma$ = 0.3 | 0.549 | -0.128 |
>
> ---
>
> **Linking theory to training noise**
> > The theory in Appendix B compares $f(\hat{c}_i)$ with $f(c^*_i)$, focusing on the error in predicting concept $i$. However, how does this prediction error relate to the _training noise rate_, which is the central focus of this paper? The setup seems to conflate inference-time uncertainty with training-time corruption–could the authors clarify this connection?
>
> Thank you for raising this point. In Appendix B we intentionally define the intervention effect at inference as $\delta_i := E_{(x, y)} [\ell(f(\hat{c}), y) - \ell(f(\hat{c}_{-i}), y)]$ to quantify how much the loss decreases when intervening on concept $i$. The key point is that $\hat{c}$ is produced by a predictor trained on noisy concept labels. Hence $\hat{c}$ already reflects the training noise rate, and so does $\delta_i$.
>
> Although the analysis is formulated at inference, it is grounded in predictions from a model affected by training-time corruption, which connects inference-time uncertainty to the training noise rate. We will make this linkage explicit in the revision.
>
> ---
>
> **Concept susceptibility and relevance**
> > The definition of susceptible concepts is based on a greater-than-average drop in accuracy under noise. But what if a concept suffers a large decline yet contributes negligibly to the final prediction (e.g., in a linear bottleneck model, if its weight is close to zero)? Shouldn’t the uncertainty-aware intervention take into account concept relevance?
>
> We appreciate the insightful observation. We agree that some concepts may suffer large accuracy declines yet have limited impact on the target. In our experiments, such cases were not frequent, but accounting for this possibility would improve diagnostic precision. As a direction for future work, we plan to incorporate measures of concept relevance into the evaluation. This extension would better capture rare cases where accuracy decline does not translate into downstream impact.
>
>
> ---
>
> References\
> [1] Shin, Sungbin, et al. "A closer look at the intervention procedure of concept bottleneck models." ICML, 2023.\
> [2] Penaloza, Emiliano, et al. "Preference optimization for concept bottleneck models." ICLR Workshop on Human-AI Coevolution, 2025.\
> [3] Sheth, Ivaxi, and Samira Ebrahimi Kahou. "Auxiliary losses for learning generalizable concept-based models." NeurIPS, 2023.

---

### Official Review · Reviewer_s86p · 2025-06-29

**Clarity:** 4
**Significance:** 3
**Originality:** 3
**Rating:** 5
**Confidence:** 4

**Summary:**

This paper presents a comprehensive study on how noisy concept annotations affect Concept Bottleneck Models (CBMs), which are designed to enhance interpretability in machine learning by predicting human-understandable concepts before making final predictions. The authors demonstrate that even moderate levels of noise in concept labels can significantly degrade CBM performance across three dimensions: predictive accuracy, interpretability, and the effectiveness of human interventions. They identify a subset of concepts that are disproportionately affected by noise and are primarily responsible for the observed performance degradation.

To understand the mechanisms behind this vulnerability, the authors analyze how noise impacts individual concept predictions and their alignment with target outputs. They find that noise does not affect all concepts equally. A small group of concepts suffers severe accuracy drops, often due to semantic ambiguity or low frequency in the dataset. These “susceptible concepts” also tend to be the most influential in determining the final prediction, meaning their corruption has an outsized impact. Furthermore, noise disrupts the alignment between concept representations and target predictions, leading to misinterpretations and reduced model reliability.

To mitigate these effects, the authors propose a two-stage framework. During training, they apply Sharpness-Aware Minimization (SAM), which encourages the model to find flatter minima in the loss landscape, thereby improving robustness to noisy labels. SAM is shown to significantly improve both concept and task accuracy, especially for the susceptible concepts. During inference, they introduce an uncertainty-guided intervention strategy that prioritizes correcting the most uncertain concept predictions. This approach leverages entropy as a proxy for susceptibility, enabling targeted corrections that restore much of the lost performance with minimal intervention.

The paper provides both theoretical and empirical support for the proposed methods. The authors derive formal justifications showing that predictive uncertainty correlates with susceptibility under reasonable assumptions. Empirical results across multiple datasets and CBM variants confirm that combining SAM with uncertainty-guided interventions consistently outperforms baseline methods. Notably, correcting just a few highly uncertain concepts can recover performance close to that of models trained on clean data, even under high noise conditions.

This study offers a principled and practical framework for enhancing the robustness of CBMs in noisy environments. It highlights the importance of identifying and protecting noise-sensitive concepts and demonstrates that targeted training and inference strategies can preserve both interpretability and predictive power. The authors also discuss limitations, such as the reliance on binary concept labels and linear predictors, and suggest future directions including extensions to hierarchical or continuous concepts and the integration of semi-supervised learning. This work lays a strong foundation for deploying interpretable models in real-world settings where annotation noise is inevitable.

**Questions:**

What are the main factors beyond ambiguity and low prevalence in the dataset that make certain concepts more susceptible?

How does SAM improve the robustness of CBMs compared to other noise mitigation techniques?

Why do you think predictive entropy is a good surrogate for susceptibility? Can you explain the intuition?

How do you think your results might generalize to other modalities?

How do you think real world noise might differ from the simulated noise in this study, and how might that alter your results?

**Ethical Concerns:**

["NO or VERY MINOR ethics concerns only"]

**Final Justification:**

I thank the authors for their additional comments and answers to my questions. My final score remains the same.

**Limitations:**

Limitations are thoughtfully considered and addressed within the paper.

**Quality:**

4

**Strengths And Weaknesses:**

Strengths:
-	This is the first comprehensive study that systematically investigates the impact of noisy concept annotations on CBMs.
-	The paper is well written with clear language and a clear articulation of the problem the authors seek to address.
-	Thoughtful experiments by the authors lead to insightful results around the subset of concepts responsible for degrading performance.
-	The mitigation strategies proposed are both theoretically sound and empirically validated.
-	The authors consider the practical implications and limitations of their work.

Weaknesses:
-	The paper only gives consideration to one modality, vision, and does not touch on generalizability to other modalities.
-	The noise is simulated and random, which may not reflect real world noise in annotations.

---

> ### Author Rebuttal · Authors · 2025-07-31
>
> We sincerely thank the reviewer for recognizing our work as insightful and comprehensive, and for providing thoughtful feedback. Below, we address the specific comments and welcome any further discussion.
>
> ---
>
> **Why SAM improves robustness**
> > How does SAM improve the robustness of CBMs compared to other noise mitigation techniques?
>
> Thank you for the insightful question. SAM optimizes by attenuating the sharpness of the loss landscape, which reduces the model’s sensitivity to noise and small perturbations. This, in turn, curbs unnecessary overfitting and promotes more stable learning under label noise. In addition, we observe that SAM tends to reduce errors specifically on susceptible concepts, providing a selective and structured mitigation where noise-induced errors concentrate.
>
> Furthermore, our theoretical analysis shows that, within the CBM structure, SAM implicitly induces an $\ell_2$-regularization effect on the final layer weights and intermediate activations, thereby lowering the sensitivity of the model to label corruption. This analysis explains why SAM yields stronger robustness in CBMs than alternatives that do not influence these internal mechanisms as directly.
>
> ---
>
> **Predictive entropy as a proxy for susceptibility**
> > Why do you think predictive entropy is a good surrogate for susceptibility? Can you explain the intuition?
>
> Thank you for the thoughtful question. The intuition is as follows. When a concept label is noisy, the corresponding concept does not present a consistent pattern in the data and thus is not learned reliably during training. The model then struggles to form a confident decision for that concept, which manifests as high predictive entropy. In contrast, informative and consistent concepts typically yield low entropy because the model can make confident predictions. Based on this intuition, we regard predictive entropy as a practical indicator of concept susceptibility.
>
> ---
>
> **Generalization across modalities**
> > How do you think your results might generalize to other modalities?
>
> Thank you for the question. CBMs are now being explored beyond vision, including settings that combine CBMs with large language models [1, 2, 3]. The core CBM process first predicts concepts and then predicts the final target, which does not depend on a specific modality. Recent LLM based CBM studies employ a similar two stage design, so we expect our main findings to transfer to these settings.
>
> That said, concept annotated datasets for LLM are not yet systematically established. An important next step is to curate well-defined concept annotated datasets suitable for LLM applications and to analyze how label noise in such datasets affects CBMs.
>
> ---
>
> **Real world noise**
> > How do you think real world noise might differ from the simulated noise in this study, and how might that alter your results?
>
> Thank you for the important question. In the main paper, we focus on uniform noise, but Appendix D.4 includes additional experiments under more realistic conditions (e.g., asymmetric, grouped noise). These experiments indicate that our core findings remain qualitatively stable.
>
> Because real world noise often depends on particular concepts and on their relation to the target, we also simulate noise with dependencies between them. Specifically, we first corrupt the target label and then add concept noise only when the target has been changed. The results below summarize the correlation between uncertainty and susceptibility under this practical setting.
>
> `Correlation between uncertainty and susceptibility on practical noise`
> | Noise Ratio | Susceptible Concepts | Non-susceptible Concepts |
> |---|---|---|
> | $\gamma$ = 0.1 | 0.646 | -0.487 |
> | $\gamma$ = 0.2 | 0.624 | -0.315 |
> | $\gamma$ = 0.3 | 0.549 | -0.128 |
>
>
> Even when noise exhibits such dependence, the positive correlation between uncertainty and susceptibility persists. We agree that considering diverse noise types is essential for broad applicability, and we appreciate the suggestion.
>
> ---
>
> **Additional factors behind susceptibility**
> > What are the main factors beyond ambiguity and low prevalence in the dataset that make certain concepts more susceptible?
>
> Thank you for the valuable question. Beyond ambiguity and low prevalence, we believe two additional properties can increase susceptibility: (1) **Context dependence.** Even for the same concept, its visual realization can vary with class, pose, or background, creating polymorphism. In such cases, the mapping from input $x$ to concept $c_i$ can become locally nonstationary, which destabilizes the decision boundary and leads to large performance drops even under mild noise; (2) **Inter-concept correlation.** When concepts are correlated, or when concept predictor $g$ shares internal representations across concepts, noise can interfere through these shared components, indirectly perturbing neighboring concepts and making their decision boundaries less stable.
>
> ---
>
> References\
> [1] Tan, Zhen, et al. “Interpreting pretrained language models via concept bottlenecks.” Pacific-Asia Conference on Knowledge Discovery and Data Mining, 2024.\
> [2] Sun, Chung-En, et al. “Concept bottleneck large language models.” ICLR, 2025.\
> [3] Bhan, Milan, et al. “Towards Achieving Concept Completeness for Textual Concept Bottleneck Models.” arXiv preprint arXiv:2502.11100 (2025).

---

> > ### Comment · Reviewer_s86p · 2025-08-04
> >
> > Thank you for the thoughtful and thorough responses to my and other reviewers' questions. I appreciate your work and wish you luck on acceptance!

---

> > > ### Author Response · Authors · 2025-08-06
> > >
> > > We sincerely thank the reviewer once again for the constructive review. We will ensure that the thoughtful discussion points (e.g., `Predictive entropy as a proxy for susceptibility`, `Generalization across modalities`, `Additional factors behind susceptibility`) are articulated more clearly in the final version, and that the additional experiment (e.g., `Real world noise`) are adequately incorporated. We are grateful for this valuable feedback.

---

### Official Review · Reviewer_uGj9 · 2025-06-30

**Clarity:** 3
**Significance:** 2
**Originality:** 1
**Rating:** 4
**Confidence:** 4

**Summary:**

This paper examines how label noise, encompassing both concept label noise and task label noise, impacts Concept Bottleneck Models (CBMs). Specifically, this work shows that (1) a CBM’s task and concept accuracies are highly susceptible to noisy label annotations, (2) concept noise is the primary source of degradation, (3) concept-label-noise-induced degradation can be attributed to small set of “susceptible” concepts, which can then be targeted during intervention to improve the effectiveness of an intervention, and (4) noise-aware losses such as Sharpness-Aware Minimization (SAM) can alleviate some of the effects of label noise. This work evaluates all these elements primarily for traditional CBMs across two datasets: CUB and AwA2. This empirical evaluation is then complemented with two theoretical results on the effect of introducing SAM to a CBM’s training and on using concept uncertainty as a proxy for identifying the set of susceptible concepts.

**Questions:**

I am currently leaning towards rejecting this work, given that I am not entirely convinced that the results presented in this work, as they are currently framed, are substantially different from those discussed in previous works. Below, I list some questions whose answers I believe could potentially add clarity or further improve this manuscript. If I have misunderstood anything at any point, please do not hesitate to let me know (sorry in advance if so!).

1. **[Significance + Originality, Critical]** I have some serious concerns regarding the novelty and significance of this paper. To help with this, it would help if you could elaborate on the following points: (1) Why are the results introduced in Section 2 not to be expected given known issues of training DNNs with noisy inputs? (2) Why are the results in Section 3.2 surprising, given that, as argued above, it is expected that in the limit, all noisy concepts will become indistinguishable? (3) Why are the SAM results in section 4.1 unexpected, given that SAM is a general-purpose optimization technique that can be applied to most DNNs, including CBMs, without much modification? (4) Why are the results in Figure 11 something unexpected, given that Shin et al. already discuss that in detail in their work? (5) How does the UCP baseline compare to the “susceptibility”-based intervention policy proposed earlier in the paper?
2. **[Quality + Clarity, Critical]** Could the authors please explain why comparisons against other relevant CBM variants or mitigation pipelines were entirely pushed to the appendix without any discussion of them in the main paper’s body? For example, if results for other potential mitigation approaches are already in the appendix, why should one not discuss them in the main body, where SAM is evaluated? Similarly, if other CBM variants have been evaluated as part of the study, why not include those results in the main body? The decision to include these in an appendix that is barely discussed in the main paper seems unusual, as they are important baselines that one would expect to have as comparisons in the main figures, not hidden in the appendices. Because of this, I believe that this work may benefit from a potentially significant rewrite of the experimentation section, focusing more on unexpected aspects.
3. **[Quality, Critical]** I have some concerns about whether SAM works in CBMs for the reasons claimed in this paper. For example, why do we not see similar improvements in robustness for other CBM variants (e.g., CEMs, ECBMs, etc) when we use SAM to train them (as seen in Table 6 of the appendix)?
4. **[Significance + Quality, Major]** Why is the goal of this work different from that of Penaloza et al. [1], as argued in Section 5? That work also seems to propose a way for mitigating label noise in CBM models. I am aware that they don’t conduct a complete study of label noise on CBMs (instead, they assume that CBMs are sensitive to this noise), but the aim to mitigate the effect of this noise seems to be shared between this paper and [1]. Given that [1] may be considered concurrent to your work, however, I only mark this as a major concern rather than a critical concern (however, at the very least, [1] should be adequately characterised in this work).
5. **[Quality, Minor]** What are the “embeddings” used in Figure 4? Are these CEM embeddings? If so, all of these important details should be immediately clear from the text.
6. **[Quality, Minor]** Why are interventions done at a per-concept level rather than at a group-level for datasets such as CUB, where there are groups of known mutually-exclusive concepts? This approach deviates from standard practices when evaluating models on this dataset, so it would benefit from further clarification.

### Minor Suggestions

Regarding suggestions for the presentation, the following are possible very minor errors and typos that could be addressed before a camera-ready version:

1. **[Nit, Potential Typo, Very Minor]** It is unclear what “it” refers to in “it first predicts…”  (line 24). If it is “CMBs”, then it should be “they” or, to make it even clearer, explicitly “CBMs” or “a CBM”.
2. **[Nit, Presentation]** The use of a circular plot in Figure 1 for representing a noise vs accuracy relationship, rather than a standard/traditional simple scatter/line plot, is a bit unusual. This may be just my personal preference, but it is much easier to read a standard scatter plot in a horizontal fashion than in a radial fashion. As such, I would reconsider the choice of plot for this figure to make the paper easier to parse and read. A similar comment applies to Figures 12 and 13 in the appendix (the former could also benefit from a larger font for the labels).

**Ethical Concerns:**

["NO or VERY MINOR ethics concerns only"]

**Final Justification:**

I have increased my score to a "Borderline accept" (4) after the rebuttal by the authors. This is because, although I believe there is value in verifying expected results do hold in practice, I am still struggling to frame most of the results in this work (except from those regarding the susceptible concept set) as entirely unexpected/novel results. Nevertheless, given that this is a well-written paper with a lot of empirical evaluations of important scenarios, I can see its value even if the core ideas are not entirely novel. As such, I am a borderline with a slight tendency towards acceptance as NeurIPS may not be the best venue for this type of work.

To further understand this reasoning, please refer to my comment in our rebuttal discussion where I outline how the authors addressed some of my concerns.

**Limitations:**

yes

**Quality:**

2

**Strengths And Weaknesses:**

Thank you for this very interesting work; I enjoyed reading it and found it easy to follow (thanks!). Below, I describe what I believe are this work’s main strengths and weaknesses:

### Strengths

1. **(S1) [Significance, Major]** This paper examines a highly popular XAI framework, namely concept bottleneck models (CBMs), from the perspective of robustness. As such, it has the potential to be of interest and use to a large subset of the NeurIPS community.
2. **(S2) [Quality, Major]** The sheer number of evaluations in the entire paper is very large, although a significant chunk of them are left entirely in the appendix (and are not discussed at all in the main body of the paper). Within those experiments discussed in the main body, the figures and tables are, for the most part, really easy to parse, read, and understand.
2. **(S3) [Originality, Minor]** Although, as argued below, this work’s originality could be stronger (as it mostly evaluates results that are to be expected given previous works), the theoretical results introduced in the appendix are interesting and new. This is particularly true for the results in Appendix B, as those in Appendix A are new but appear to be a simple readjustment of previous work by Baek et al.
4. **(S4) [Clarity, Minor]** The paper is very clearly written and well-organized.

### Weaknesses

1. **(W1) [Originality + Significance, Critical]** Although it is always important to verify expected claims empirically and theoretically, in my opinion, most of the results presented in this work are to be expected based on well-known limitations of DNNs (not just CBMs). For example, if dataset noise increases, it is well-known and expected that we will struggle to learn a classifier (which, if used as the input of yet another classifier, as in CBMs, that new combined downstream classifier will also struggle). Similarly, if noise increases on features such as concepts, then it is expected that, in the limit, all concepts will become indistinguishable from each other.  Therefore, as noise increases, it is not unexpected that all concepts become equally (although spuriously) important to any model trained on them (as seen in Section 3). Similar points apply to the other observations/results of this paper (including intervention results and the mitigating experiments, which use an already-existing mitigation approach for DNNs to improve robustness in CBMs). Because of this, I have some hesitations about the potential novelty and contribution of this work vis-à-vis already well-established results.
2. **(W2) [Quality, Critical]** The study of label noise in the main body of the paper is restricted uniquely to traditional CBMs. This leaves aside a wealth of better-performing, highly popular methods that have emerged since, including CEMs, ProbCBMs, ECBMs, Post-hoc CBMs, Label-free CBMs, Stochastic CBMs, and others. These are only discussed in the appendices. This seems like an odd decision given that they are important considerations. More importantly, however, looking at the SAM results for these baselines casts some doubts on whether the results improvements from SAM observed in CBMs in the main body of the paper properly generalize. Please see below for specific questions regarding this.
3. **(W3) [Quality, Critical]** The evaluation of the proposed mitigation approaches is lacking a non-trivial evaluation baseline in the main body (i.e., not discussed in the appendix). This is particularly important if the main contribution for this mitigation pipeline is applying an already existing method to CBMs (if that's the case, then the novelty could be in explaining WHY one should use SAM over any of the other existing approaches to deal with label noise). Moreover, the SAM experiments would significantly benefit from explaining how SAM compares against recently proposed concept-specific methods that aim to handle the exact same problem (e.g., [1], although I am aware this paper might've come concurrently with this work). Similarly, the concept intervention experiments for the proposed policy in Figure 9 do not include any non-trivial intervention policies (e.g., Coop [1] or UCP and CCTP). These are later discussed in Figure 11, but they appear to be discussed as if these results were previously unknown. However, the efficacy of UCP w.r.t. a random baseline and CCTP was well-known from the paper that introduced both. What would be more relevant for this paper is to see how it compares to the policy “susceptibility”-based policy introduced here.
4. **(W4) [Clarity, Major]** Many key results are included and discussed only in the appendix (e.g., comparisons for different CBMs, against other mitigation strategies, etc). Yet, several of these appendix experiments are critical and more important than those expected from previous works, such as those in Section 2 and Figure 11. Therefore, I would strongly suggest that the authors reconsider the structure and discussion of these results in the main paper. At the very least, the main takeaway of the results discussed in the appendices should be included in the main paper (currently, there is no easy way to navigate the appendices other than reading them end-to-end, as there are no mentions of the main messages of each appendix anywhere in the main body of the paper).
5. **(W5) [Quality, Major]** Some of the experiments, for example, the results summarized in Figure 1 and the visualizations reported in Figure 4, are hard to interpret or are missing key details. See below for specific questions.

### References

1. Penaloza et al. Preference optimization for concept bottleneck models. In ICLR Workshop on Human-AI Coevolution, 2025.
2. Chauhan et al. "Interactive concept bottleneck models." *Proceedings of the AAAI Conference on Artificial Intelligence*. Vol. 37. No. 5. 2023.

---

> ### Author Rebuttal · Authors · 2025-07-31
>
> We sincerely thank the reviewer for the thoughtful, constructive, and detailed feedback. We address the reviewer’s specific comments below and welcome any further discussion.
>
> ---
>
> **Unexpected findings**
>
> We appreciate the reviewer’s point. This work, however, presents results that were not previously documented and are difficult to anticipate.
>
> First, it is commonly believed that CBMs allow one to correct the final prediction via intervention even when a concept is mispredicted. From this view, one might expect that models trained with noisy concepts would readily recover performance through intervention. Our experiments contradict this expectation. As shown in Section 2.4, when CBMs are trained on datasets with concept noise, the intervention effectiveness itself is substantially reduced, so the expected recovery does not materialize. Given that real deployments often rely on concept sources with nontrivial noise, this indicates that intervention may fail to operate as intended and can lead to unexpected failure.
>
> Second, CBMs degrade in a qualitatively different manner from standard DNNs under the same noise level. For example, on the CUB classification task with 0%, 20%, and 40% noise, a DNN exhibits a relatively moderate decline (79.7 → 65.9 → 53.0), whereas a CBM shows an abrupt collapse (74.3 → 50.3 → 4.0). This occurs because the two-stage structure of CBM propagates errors in concept prediction directly to the target predictor. The evidence shows that CBMs are considerably more vulnerable to concept noise than is typically assumed.
>
> These findings reveal an inherent sensitivity of CBMs to concept noise and underscore the need to recognize this vulnerability and develop appropriate mitigation techniques in order to build reliable CBMs.
>
> ---
>
> **Significance of our work**
>
> We appreciate the constructive comment. The significance of our study can be summarized in three parts.
>
> * To our knowledge, this is the first systematic analysis of the noisy concept annotation problem in CBMs, a practical issue that can arise from human error and subjective judgment yet has remained largely unattended. We show, through multi-faceted analyses, that such noise negatively affects all core CBM properties: interpretability, intervention effectiveness, and predictive performance.
>
> * We find that degradation is not uniform across concepts. Rather, it concentrates on a subset that is **susceptible** to noise. This small set explains most of the accuracy decline and therefore drives the overall performance drop.
>
> * To mitigate the risks posed by susceptible concepts, we propose sharpness-aware minimization during training and uncertainty-guided intervention at inference. We provide broad empirical results and theoretical support demonstrating that these strategies are effective at restoring the reliability of CBMs.
>
> ---
>
> **Why SAM**
>
> We understand the reviewer’s concerns. We adopted SAM because prior work by Baek et al. [1] indicates that SAM is a representative and often effective approach for alleviating label noise, and it can outperform alternative methods [2, 3, 4]. In our experiments, SAM also handled noisy concept annotations in CBMs, especially for susceptible concepts. As noted in Appendix D.3, we do not claim SAM is the only solution; rather, for the purposes of this work, we present it as one effective approach to the observed problem.
>
> ---
>
> **Compare to [5]**
>
> Thank you for the comment. We clarify the differences from the concurrent work [5]. While [5] shares the motivation that CBMs are sensitive to label noise, it primarily focuses on improving the learning objective, assuming this sensitivity without further investigation. It does not provide a comprehensive analysis of how noise affects all key CBM components (e.g., interpretability, intervention effectiveness, and predictive performance). In contrast, our work quantifies noise effects across these core properties, identifies their causes, and proposes an integrated mitigation framework spanning both training and inference. We will make this distinction clearer in the main text.
>
> We also conducted additional comparisons between SAM and CPO from [5], which modifies the loss from a preference optimization perspective and focuses primarily on joint CBM training, with sequential training as a secondary case. We report comparisons between them in the table below.
>
> `Concept Accuracy`
> |Model|Type|0%|10%|20%|30%|40%|Avg|
> |-|-|-|-|-|-|-|-|
> |CBM(seq)|Vanilla|94.9|89.0|81.0|71.1|60.6|79.3|
> |“|CPO|95.5|89.9|80.6|70.3|60.0|79.3|
> |“|SAM|95.3|90.0|82.4|**72.3**|**61.3**|80.3|
> |“|CPO+SAM|**95.7**|**91.3**|**82.7**|**72.3**|60.7|**80.5**|
> |CBM(joint)|Vanilla|94.6|88.5|80.3|70.4|60.2|78.8|
> |“|CPO|70.5|64.5|61.9|58.8|53.9|61.9|
> |“|SAM|**95.2**|**89.4**|**81.6**|**71.5**|**60.9**|**79.7**|
> |“|CPO+SAM|70.8|66.3|63.0|58.9|53.8|62.6|
>
> `Task Accuracy`
> |Model|Type|0%|10%|20%|30%|40%|Avg|
> |-|-|-|-|-|-|-|-|
> |CBM(seq)|Vanilla|66.5|48.2|33.8|17.5|5.0|34.2|
> |“|CPO|68.8|51.9|32.9|15.0|5.0|34.7|
> |“|SAM|69.0|52.1|**37.5**|**20.0**|**6.1**|**36.9**|
> |“|CPO+SAM|**69.5**|**54.9**|36.4|18.5|5.4|**36.9**|
> |CBM(joint)|Vanilla|66.0|52.2|34.8|16.2|4.6|34.8|
> |“|CPO|70.6|63.5|55.8|47.3|36.1|54.7|
> |“|SAM|69.9|54.5|37.0|16.8|5.0|36.6|
> |“|CPO+SAM|**73.7**|**67.3**|**60.0**|**51.1**|**40.6**|**58.5**|
>
> The main takeaways are: (1) In joint training, CPO tends to yield higher target accuracy, while SAM tends to preserve concept accuracy better; (2) In sequential training, SAM generally outperforms CPO, and combining the two can sometimes achieve the highest performance. These results suggest that two approaches are complementary and potentially usable together.
>
> ---
>
> **On Appendix materials**
>
> Thank you for the helpful comments. We placed the label noise mitigation experiments in the Appendix because the alternative methods did not achieve improvements comparable to SAM, so we treated them as supporting results. We referenced these analyses in the main conclusion under “further analyses”, and in the final version we will make the connection to Section 4.1 explicit.
>
> We also placed SAM on CBM variants experiments in the Appendix because they showed trends similar to the main results and due to space constraints. In light of your comments, we revisited these analyses and added further evaluations across CBM variants. In particular, we examined the effect of SAM on susceptible concepts. For SCBM, SAM showed clear improvements over SGD: at 20% noise, 75.8 vs. 74.9 (+0.9), and at 40% noise, 60.1 vs. 56.4 (+3.7). For AR-CBM and CEM, the gains were smaller: AR-CBM at 20% noise 71.9 vs. 71.3; at 40%, both 53.9; CEM at 20%, 73.3 vs. 73.1; at 40%, 54.6 vs. 54.5.
>
> These results indicate that the effect of SAM differs across CBM variants, and that resilience on susceptible concepts may depend on model design. While our study centers on CBMs, we agree that a systematic analysis across variants is important; we will reflect these points in the main text and consider a broader framework that covers such generalization in future work.
>
> ---
>
> **Uncertainty-guided intervention**
>
> Thank you for the comment. There seems to be some confusion regarding Figure 11. Our goal there is not to newly propose uncertainty guided intervention, but to analyze why this strategy becomes more effective when label noise is present. We state this explicitly in the paper. Our new observation is that the relative improvement of uncertainty-guided intervention increases as noise grows, a point that was not detailed in [6].
>
> The core message of Figure 9 is that prioritizing interventions by susceptibility is effective. To make this clear, we compare the susceptibility-based policy not only with a random policy but also with stronger baselines. The comparative tables are included below.
>
> `Noise 10%`
> |Task Acc.|0|1|2|3|4|5|
> |---|---|---|---|---|---|---|
> |Susceptibility|61.9|**69.2**|**74.8**|**79.7**|**83.7**|**86.1**|
> |UCP|61.9|66.1|69.3|72.4|75.3|77.6|
> |CCTP|61.9|62.3|62.9|63.7|64.8|66.3|
> |Random|61.9|62.8|63.5|64.4|65.1|65.9|
>
> `Noise 20%`
> |Task Acc.|0|1|2|3|4|5|
> |---|---|---|---|---|---|---|
> |Susceptibility|51.9|**65.3**|**75.0**|**82.0**|**86.6**|**89.4**|
> |UCP|51.9|57.7|63.1|68.0|71.8|75.3|
> |CCTP|51.9|52.8|54.1|55.8|57.9|59.8|
> |Random|51.9|53.2|54.8|55.9|57.4|58.7|
>
> `Noise 30%`
> |Task Acc.|0|1|2|3|4|5|
> |---|---|---|---|---|---|---|
> |Susceptibility|33.1|**47.5**|**60.8**|**71.3**|**78.7**|**83.3**|
> |UCP|33.1|38.4|44.0|49.6|54.3|58.9|
> |CCTP|33.1|34.2|35.7|37.9|40.9|43.7|
> |Random|33.1|34.3|35.9|37.6|38.9|40.5|
>
> These results show that prioritizing susceptible concepts remains an efficient intervention policy even against non‑trivial strategies such as UCP and CCTP. Accordingly, we believe the validity of our core intervention strategy is well supported.
>
> ---
>
> **Minor points**
>
> We appreciate the careful, detailed feedback. We will correct the noted typographical issues and revise figures to improve readability in the final version. The embeddings in Figure 4 refer to the representations produced by the concept predictor $g$. Interventions are conducted at the individual concept level to analyze the effect of each concept on prediction through controlled intervention. If group-level evaluations are preferred, we are happy to add those experiments upon request.
>
> ---
>
> References\
> [1] Baek, Christina, Zico Kolter, and Aditi Raghunathan. "Why is SAM robust to label noise?" ICLR, 2024.\
> [2] Zhang, Hongyi, et al. "mixup: Beyond empirical risk minimization." ICLR, 2018.\
> [3] Arazo, Eric, et al. "Unsupervised label noise modeling and loss correction." ICML, 2019.\
> [4] Jiang, Lu, et al. "Mentornet: Learning data-driven curriculum for very deep neural networks on corrupted labels." ICML, 2018.\
> [5] Penaloza, Emiliano, et al. "Preference optimization for concept bottleneck models." ICLR Workshop on HAIC, 2025.\
> [6] Shin, Sungbin, et al. "A closer look at the intervention procedure of concept bottleneck models." ICML, 2023.

---

> > ### Comment · Reviewer_uGj9 · 2025-08-04
> >
> > Thank you so much for taking the time to carefully read over my review and write this rebuttal. Below I comment on some of the points made above:
> >
> > ### > Unexpected Findings
> >
> > The rebuttal claims that "*... it is commonly believed that CBMs allow one to correct the final prediction via intervention even when a concept is mispredicted. From this view, one might expect that models trained with noisy concepts would readily recover performance through intervention.*" However, I strongly disagree with this statement. I would argue that, very likely, the authors would agree with me that if all concept labels are noise, then interventions should surely not work. Therefore, if you think of the transition from perfect ground-truth concept labels to entirely random labels as a continuum, then it is not unexpected at all that somewhere in the middle of that continuum interventions will stop working well, getting only worse and worse as the concept labels are noisier and noiser. Hence, I would argue that this is very much expected. What would be surprising, say, is if the transition between interventions "working" and not "working" as noise increases is non-continuous (something I would not expect to be the case at all).
> >
> > Similarly, I would still argue that the compounding nature of errors from the label predictor is to be expected. However, I can see that quantifying the extent of this effect empirically can be an interesting new addition.
> >
> > ### > Significance of our work
> >
> > I agree with the first two points of this answer. However, I believe the last point is a bit trickier as the paper applies already existing methods (e.g. intervention policies and SAM) that were designed to exactly already address some of the issues brought up earlier in the paper. Nevertheless, my main issue with the novelty of this work is the lack of unexpected results. Unfortunately, besides the finding of the susceptible concept set and the susceptibility policy derived from those, I am struggling to see the other contributions as unexpected. Again, I want to emphasise that this does not mean these studies are not good or needed. However, it does mean that they are a big weakness to point out for a conference that focuses mostly on novel/unexpected/entirely new results.
> >
> > ### Other New Results (comparisons against other mitigation and intervention baselines)
> >
> > Thanks for these new results! These are interesting and worthwhile new inclusions! I would strongly suggest to include these results as part of the main body camera-ready paper if this work gets accepted (rather than in a new appendix).
> >
> >
> > ## Updated Review
> >
> > Given all the new results and discussion above, **I will increase my score to a weak reject (3)**. This is because, although I do believe there is significant value in verifying expected effects, I am still not entirely convinced that this paper's contributions are unexpected/novel to the point of what is usually expected of a NeurIPS papers. Nevertheless, I am happy to be convinced otherwise by the AC or my fellow reviewers, in which case I am happy to increase my score even further.
> >
> > I wish the authors the best of luck with this submission and thank them for carefully going over my feedback.

---

> > > ### Author Response · Authors · 2025-08-06
> > >
> > > We appreciate the reviewer’s careful reading, constructive discussion of our additional experiments and revisions, and the increase in score. This feedback has been invaluable in strengthening our work. We address the comments below and welcome further discussion.
> > >
> > > ---
> > >
> > > **On what counts as _unexpected_**
> > >
> > > It is not obvious that our findings should be deemed _expected_. Prior work [1] shows that deep networks can learn from data in which true labels are vastly outnumbered by incorrect labels, indicating that label noise does not necessarily lead to straightforward degradation in prediction accuracy. By analogy, one cannot a priori assume that making concept labels noisier will always worsen a CBM’s concept predictions in a simple, monotonic way. Moreover, because CBMs employ a two-step prediction pipeline, the effect of concept noise cannot be asserted with certainty without actually seeing it. In fact, as we reported earlier, performance under noise is relatively robust for standard DNNs, whereas CBMs exhibit markedly larger drops.
> > >
> > > With respect to intervention, there are also genuinely unknown aspects. For example, in our experiments we observed a gradual decline in post-intervention accuracy as noise increased from 0% to 30% (99.8% → 98.4% → 95.7%). By contrast, in another configuration the accuracy fell sharply between 30% and 40% (89.4% → 53.1%). Given the reviewer’s remark that a non-continuous transition would be surprising, these results quantitatively show that the shift from “working” to “not working” need not be smooth.
> > >
> > > Beyond this, we report several new observations: the identification of a susceptible concept set; evidence that SAM mitigates errors more effectively on this susceptible set than on the non-susceptible set; that intervening on the susceptible set yields larger performance gains than advanced intervention policies; and that uncertainty-guided intervention serves as a practical proxy for susceptibility in real settings. To our knowledge, these have not been documented before and they provide actionable guidance for CBM practice. We believe these findings provide practical value by empirically testing and refining intuitions about CBMs, thereby informing how to assess and improve their reliability.
> > >
> > > ---
> > >
> > > **Why confirming _expected_ effects still matters**
> > >
> > > We appreciate the reviewer’s agreement with two of our three contributions and would like to clarify the remaining point.
> > >
> > > The fact that some effects may seem anticipated does not diminish the value of a systematic, quantitative analysis. While our mitigation strategies (e.g., SAM and uncertainty-guided intervention) build on existing methods and may not be algorithmically novel, our contribution is to demonstrate that they are effective precisely for noise-susceptible concepts, thereby improving the practical reliability of CBMs.
> > >
> > > We view this work as a first step; it analyzes how concept noise affects CBMs and proposes a concrete, empirically supported mitigation direction. We expect more sophisticated approaches to build on this foundation. Confirming effects that might appear _expected_ is neither trivial nor optional when the goal is to establish reliable methodology.
> > >
> > > ---
> > >
> > > **On the additional comparisons**
> > >
> > > Thank you for the positive assessment of the new comparisons. As suggested, we will seriously consider incorporating these results as part of the main body in the final version. We agree they are important for showing how our analysis framework and mitigation approach relate to existing alternatives, and we appreciate the constructive recommendation.
> > >
> > > ---
> > >
> > > Thank you again for the thoughtful feedback and engagement. Your comments helped us clarify our contributions and sharpen the presentation, and we are grateful for the substantive scholarly exchange.
> > >
> > > ---
> > >
> > > References\
> > > [1] Rolnick, David, et al. "Deep learning is robust to massive label noise." arXiv preprint arXiv:1705.10694 (2017).

---

> > > > ### Comment · Reviewer_uGj9 · 2025-08-06
> > > >
> > > > Thank you so much for the follow-up and the continuous discussion. After carefully reading this reply, as well as my fellow reviewers' replies, I have decided to **increase my score to a borderline accept**. This is because, although I do see value in this work, I am still not 100% convinced of its suitability for NeurIPS, given the lack of, in my opinion, a key "driver" novel/unexpected result.
> > > >
> > > > I wish the authors the best of luck with this submission and thank them for their discussion and time!

---

> > > > > ### Author Response · Authors · 2025-08-08
> > > > >
> > > > > We are grateful for the insightful engagement and constructive feedback, which have sharpened the core arguments and strengthened the overall contribution of our study. We sincerely appreciate your time and effort, as well as the generous increase in score. In the final version, we will articulate the principal discussion areas more explicitly (e.g., `Unexpected findings`, `Significance of our work`, `Why SAM`, `On Appendix materials`, `Minor points`) and incorporate the additional experiments as recommended (e.g., `Compare to [5]`, `Uncertainty-guided intervention`).

---

### Official Review · Reviewer_B7SW · 2025-07-03

**Clarity:** 2
**Significance:** 3
**Originality:** 3
**Rating:** 5
**Confidence:** 4

**Summary:**

This work investigates how label noise corrupts concept bottleneck models (CBMs), degrading their prediction accuracy, interpretability, and intervention utility. The authors identify that performance loss is caused by a vulnerable subset of concepts, those most sensitive to annotation errors. To address this, the authors introduce (1) sharpness-aware training to stabilize concept learning and (2) inference-time entropy ranking to detect and correct only the most noise susceptible concepts. The framework preserves CBM reliability under noise while maintaining full interpretability, supported by theoretical analysis and empirical validation.

**Questions:**

1. If only a small subset of concepts is susceptible noise, does this limit the practical impact of label noise on CBMs if they do not occur on susceptible noise?
2. Figure 10 shows unclear separation between susceptible and non-susceptible concepts. Are these patterns consistent across different datasets. How many datasets have the authors evaluate?
3. Is noise applied independently to each concept or the entire dataset? This significantly affects interpretation of the results.
4. For globally applied noise, does the model consistently identify the same concepts as susceptible across different noise instantiations, or does susceptibility vary randomly? I suggest the authors track these susceptible concepts.
5. Target noise does not affect the performance, even at 40% corruption, which seems odd. Why?
6. In Figure 7, why do many initially low frequency concepts show frequency rise (2000x+) after noise injection?

**Ethical Concerns:**

["NO or VERY MINOR ethics concerns only"]

**Final Justification:**

The authors have provided a thorough response that addresses most of my concerns. As my primary issues have been resolved, I maintain my original score, 5 Accept.

**Limitations:**

Yes

**Quality:**

3

**Strengths And Weaknesses:**

Strengths:
1. Investigating noisy labels in CBMs presents a meaningful study to learning with noisy labels.
2. The work conducts extensive and thoughtful experiments, thoroughly analyzing the impact of noise across multiple dimensions.

Weaknesses:
1. Since SAM improves performance even on clean data, its specific role in mitigating noisy impacts remains unclear.
2. The noisy impact is highly dependent on hyperparameters (e.g., epochs, learning rate) and noise type. These settings should be explicitly clarified in Sections 2–3 for reproducibility.
3. Experiments were run across three different GPUs, making replication difficult. At minimum, the same hardware should be used for comparable experiments, and results should specify the GPU used.
4. The observed noise effects resemble Subclass-Dominant Label Noise (SDN) [1], where intra-class attributes influence robustness. A direct comparison would strengthen the analysis.

[1] Subclass-Dominant Label Noise: A Counterexample for the Success of Early Stopping

---

> ### Author Rebuttal · Authors · 2025-07-31
>
> We sincerely thank the reviewer for recognizing the value of our work and providing constructive feedback. We address the reviewer’s specific comments below and welcome any further discussion.
>
> ---
>
> **Role of SAM under clean and noisy conditions**
> > Since SAM improves performance even on clean data, its specific role in mitigating noisy impacts remains unclear.
>
>
> Thank you for this observation. SAM is known to enhance generalization even in clean settings, and we observe a similar effect for CBMs. Our primary focus, however, is not merely the generalization benefit of SAM but how SAM specifically mitigates label noise within CBMs. In particular, as shown in Table 2, we find that SAM tends to reduce prediction errors concentrated on **susceptible** concepts, i.e., those most sensitive to annotation errors. This indicates a selective, structural mitigation effect beyond an overall accuracy increase.
>
>
> ---
>
>
> **Reproducibility**
> > The noisy impact is highly dependent on hyperparameters and noise type (e.g., epochs, learning rate). These settings should be explicitly clarified in Sections 2-3 for reproducibility.
> > Experiments were run across three different GPUs, making replication difficult. At minimum, the same hardware should be used for comparable experiments, and results should specify the GPU used.
>
>
> Thank you for the thoughtful feedback. Most hyperparameters (e.g., epochs, learning rates) are specified in Appendix C.3. We agree that placing the essential settings in the main text would aid readers, and we will move the relevant details into Sections 2-3 in the final version.
>
>
> Regarding hardware, the majority of experiments were run on NVIDIA RTX 3090, with some on A6000 and A100 due to compute availability. Although not all comparisons were executed on identical hardware, all comparative experiments were performed under same conditions (e.g., random seeds and hyperparameters), so we believe that hardware differences do not materially affect the conclusions. To further strengthen reproducibility, we will re-run the experiments that were not on the 3090 so that comparable results are consolidated on the same GPU.
>
>
> ---
>
>
> **Comparison with SDN**
> > The observed noise effects resemble Subclass-Dominant Label Noise (SDN), where intra-class attributes influence robustness. A direct comparison would strength the analysis.
>
>
> We appreciate the pointer to this closely related work. There are clear parallels. For example, our t-SNE visualizations show entanglement across classes under noise, qualitatively similar to the observations of SDN. In addition, the noise setting of SDN is related to our grouped noise setting (Appendix C.4 and D.4). Interpreting the subclasses of SDN as units analogous to our concepts makes the similarity more apparent.
>
>
> A key difference is scope and objective. SDN focuses on a case where early stopping fails to alleviate label noise in DNNs and proposes a remedy. Our study quantitatively analyzes how concept noise affects interpretability, intervention effectiveness, and prediction accuracy in CBMs, and proposes an integrated mitigation framework spanning both training (SAM) and inference (uncertainty-guided intervention). We will incorporate an explicit discussion of SDN in the final version. Thank you for the valuable suggestion.
>
>
> ---
>
>
> **Impact of label noise on susceptible sets**
> > If only a small subset of concepts is susceptible noise, does this limit the practical impact of label noise on CBMs if they do not occur on susceptible noise?
>
>
> Thank you for the question. If noise does not occur on the susceptible set, the overall impact on a CBM can indeed be limited. While non-susceptible concepts may also experience some degradation, our results indicate that the aggregate performance drop is largely driven by noise on the susceptible concepts. Thus, we expect the effect of noise is substantially reduced when it does not target them.
>
>
> ---
>
>
> **Susceptibility across datasets**
> > Figure 10 shows unclear separation between susceptible and non-susceptible concepts. Are these patterns consistent across different datasets. How many datasets have the authors evaluate?
>
>
> Thank you for raising this point. While the main paper analyzes susceptibility on CUB, we agree it is important to examine whether the pattern generalizes. We therefore conducted the same analysis on AwA2 as an additional experiment; results are summarized below.
>
>
> `Correlation between uncertainty and susceptibility on AwA2`
> | Noise Ratio | Susceptible Concepts | Non-susceptible Concepts |
> |---|---|---|
> | $\gamma$ = 0.1 | 0.540 | -0.560 |
> | $\gamma$ = 0.2 | 0.355 | -0.382 |
> | $\gamma$ = 0.3 | 0.175 | -0.167 |
>
>
> These results suggest that a positive correlation between uncertainty and susceptibility persists across different datasets. While the AwA2 dataset exhibits a clear negative correlation within the non-susceptible set, this does not undermine the positive relationship observed in the susceptible set. This extension supports that our analysis captures a more general pattern.
>
>
> ---
>
>
> **Noise injection protocol**
> > Is noise applied independently to each concept or the entire dataset? This significantly affects interpretation of the results.
> > For globally applied noise, does the model consistently identify the same concepts as susceptible across different noise instantiations, or does susceptibility vary randomly? I suggest the authors track these susceptible concepts.
>
>
> We appreciate the question and suggestion. In the main paper, noise is applied at the concept level. Also, the susceptible concepts identified for each random seed are reported below.
>
>
> `Susceptible concepts across seeds`
> | Seed | Susceptible Concepts (Unsorted) |
> |---|---|
> | Seed 1 | 4, 7, 9, 10, 13, 20, 21, 23, 25, 28, 32, 39, 44, 53, 56, 58, 76, 77, 89, 91, 96, 102, 104, 107 |
> | Seed 2 | 4, 7, 10, 13, 20, 21, 23, 25, 28, 30, 32, 53, 56, 58, 75, 76, 77, 89, 91, 96, 102, 107 |
> | Seed 3 | 4, 7, 10, 13, 20, 21, 25, 28, 30, 36, 53, 56, 58, 63, 70, 75, 76, 77, 91, 96, 99, 107 |
> | Intersection | 4, 7, 10, 13, 20, 21, 25, 28, 53, 56, 58, 76, 77, 91, 96, 107 |
>
>
> We observe that similar concepts are repeatedly identified as susceptible across seeds, suggesting structural sensitivity rather than random fluctuation.
>
>
> ---
>
>
> **Why target noise appears benign**
> > Target noise does not affect the performance, even at 40% corruption, which seems odd. Why?
>
>
> Thank you for raising this. In our CBM configuration, the target predictor $f$ follows prior work [1] and is a single linear layer from concepts to the target. Its limited capacity makes it less able to overfit noisy target labels, reducing sensitivity to target corruption. Moreover, CBM training primarily burdens the concept predictor $g$. When $g$ produces stable, accurate concept representations, $f$ operates on relatively clean inputs, so even substantial target noise has a muted effect on final accuracy. We will expand our discussion of this phenomenon in the final version.
>
>
> ---
>
>
> **Concept frequency shift under noise**
> > In Figure 7, why do many initially low frequency concepts show frequency rise (2000x+) after noise injection?
>
>
> Thank you for the question. Before noise injection, the dataset has a favorable signal-to-noise ratio and some concepts appear very rarely. Under uniform concept noise, each concept value is flipped with a fixed probability, which increases the chance that originally rare concepts appear due to random perturbations. As a result, low-frequency concepts can experience sharp relative increases in frequency. We view this as a distributional shift induced at the data-level by the noise process rather than a change in the model’s intrinsic prediction tendency.
>
>
> ---
>
> References\
> [1] Koh, Pang Wei, et al. "Concept bottleneck models." ICML, 2020.

---

> > ### Comment · Reviewer_B7SW · 2025-08-04
> >
> > The authors have provided a thorough response that addresses most of my concerns. As my primary issues have been resolved, I maintain my original score, 5 Accept.

---

> > > ### Author Response · Authors · 2025-08-06
> > >
> > > We are pleased that our response adequately addresses the concerns raised. We will incorporate into the final version the discussion points identified by the reviewer (e.g., `Role of SAM under clean and noisy conditions`, `Comparison with SDN`, `Why target noise appears benign`, `Concept frequency shift under noise`) as well as relevant experiments (e.g., `Reproducibility`, `Susceptibility across datasets`, `Noise injection protocol`). We sincerely appreciate this constructive feedback.

---

### Decision · Program_Chairs · 2025-09-17

**Decision:**

Accept (poster)

**Comment:**

This work examines how label noise, encompassing both concept label noise and task label noise, impacts Concept Bottleneck Models. Specifically, this work shows that a CBM’s task and concept accuracies are highly susceptible to noisy label annotations,  concept noise is the primary source of degradation, and concept-label-noise-induced degradation can be attributed to small set of “susceptible” concepts, which can then be targeted during intervention to improve the effectiveness of an intervention, and (4) noise-aware losses such as Sharpness-Aware Minimization (SAM) can alleviate some of the effects of label noise.

The reviewers agreed after some rebuttal time that this is a well-written and timely paper that provides a useful empirical analysis of CBMs under noisy supervision and should be accepted to NeurIPS. In the final version, the authors should provide a clearer explanation and additional experiments regarding "On the independence assumption", as suggested by the reviewers, to improve the narrative.